# SHADOW-FT: TUNING INSTRUCT MODEL VIA TRAINING ON PAIRED BASE MODEL

## ABSTRACT

Large language models (LLMs) consistently benefit from further fine-tuning on various tasks. However, we observe that directly tuning the INSTRUCT (i.e., instruction tuned) models often leads to marginal improvements and even performance degeneration. Notably, paired BASE models, the foundation for these INSTRUCT variants, contain highly similar weight values (i.e., less than 2% on average for Llama 3.1 8B). The BASE model tends to be a good learner yet a weak backbone without post-training. Therefore, we propose a novel **Shadow-FT** framework to tune the INSTRUCT models by leveraging corresponding BASE models. The key insight is to fine-tune the BASE model, and then *directly* graft the learned weight updates to the INSTRUCT model. Our proposed Shadow-FT introduces no additional parameters, is easy to implement, and significantly improves performance. We conduct extensive experiments on tuning mainstream LLMs, such as Qwen 3 and Llama 3 series, and evaluate them across 19 benchmarks covering coding, reasoning, and mathematical tasks. Experimental results demonstrate that Shadow-FT consistently outperforms conventional full-parameter and parameter-efficient tuning approaches. Further analyses indicate that Shadow-FT can be applied to multimodal large language models (MLLMs) and combined with direct preference optimization (DPO).

## 1 INTRODUCTION

Large Language Models (LLMs), such as Qwen (Bai et al., 2023), Llama (AI@Meta, 2024), and Gemma (Team et al., 2025a), have demonstrated remarkable performance across diverse disciplines (Zhang et al., 2023; Wang et al., 2024a). Such a strong capability is always attributed to the pre-training on massive data with billions of parameters (Bi et al., 2024; Tao et al., 2024). When applied in real-world scenarios, there are several challenges. The users want the LLMs to follow their instructions helpfully and honestly (Li et al., 2024), which is not covered during the pre-training (Zhang et al., 2023; Liu et al., 2024). Meanwhile, the downstream tasks always involve specific domain knowledge requiring adaptation (Wang et al., 2023; Luo et al., 2024).

To tackle these issues, one predominant approach is further tuning LLMs on desired tasks, including full parameter fine-tuning and parameter-efficient fine-tuning (Liu et al., 2021; Hu et al., 2022). Typically, for each model size, two paired variants are provided: the pretrained base model (denoted as BASE) and its instruction-tuned version (denoted as INSTRUCT). The BASE model exhibits relatively poor instruction-following ability (i.e., *a weak backbone*), while the INSTRUCT model performs better. However, we observe that tuning the INSTRUCT models brings marginal improvements and even a performance degeneration. Therefore, how to tune the INSTRUCT model effectively gains increasing importance.

In this paper, we first analyze the weights of paired BASE and INSTRUCT models considering the relative absolute difference $\sigma$. Fortunately, we find that the weights of BASE and INSTRUCT are highly similar. As shown in Figure 1, the gap $\sigma$ is quite low, with an average $\sigma$ of 0.016 for the Llama-3.1-8B model. Intuitively, the contained instruction-following ability of INSTRUCT model disturbs the learning of new knowledge, while BASE can avoid it. We further provide a deep analysis to prove this conclusion. Motivated by these, we thus propose a novel **Shadow-FT** framework to employ the BASE model as 'shadow' of INSTRUCT. The key is to tune the BASE for better weight updates and directly graft these updates to INSTRUCT, as they share the same structures.

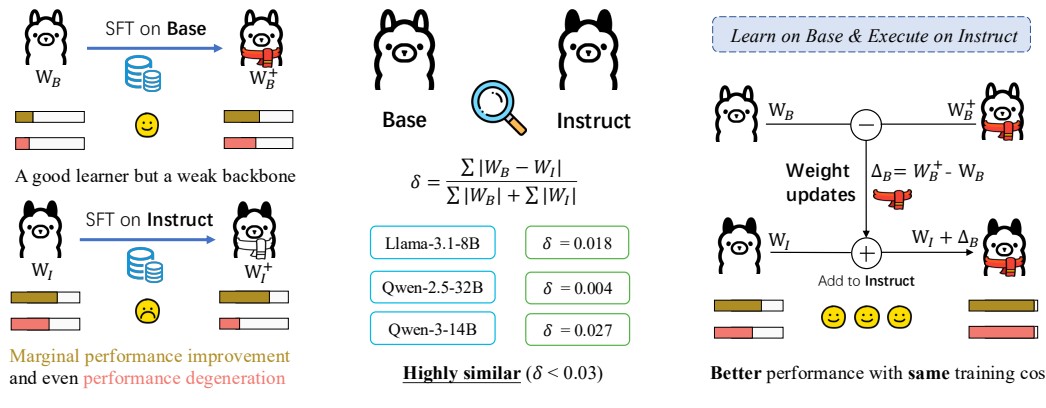

Figure 1: Performance of vanilla SFT (part a), similarity on weights(part b), and the Shadow-FT framework (part c). The progress bars in brown and pink denote the different abilities, **the fuller, the better**. Based on the SFT dynamics and weight similarity (gap $\sigma$ less than 0.03), we propose to tune the paired BASE model and then graft the weight updates onto INSTRUCT model.

To evaluate the performance, we conduct extensive experiments tuning mainstream LLMs such as Qwen 3 (Bai et al., 2023) and Llama 3 (AI@Meta, 2024). For the tuning data, we employ the BAAI-Infinity-Instruct Dataset[1] and extract 2000 samples named as BAAI-2k following (Zhou et al., 2023; Muennighoff et al., 2025). Without the loss of generality, we apply Shadow-FT on full parameter and low-rank settings, and then report the performance on 19 datasets. Experimental results indicate that Shadow-FT consistently outperforms the baselines under various settings, demonstrating its effectiveness and robustness. Further analyses show that Shadow-FT can be applied to MLLMs and combined with DPO for alignment. Our contributions can be concluded as follows:

- We find that paired BASE and INSTRUCT are highly similar considering weight values, and thus propose a novel Shadow-FT framework. The key is to tune the BASE for better weight updates and directly graft these updates to INSTRUCT.

- We conduct extensive experiments tuning various mainstream LLMs and report the performance on 19 benchmarks across math, code, and reasoning. Experimental results demonstrate the effectiveness and robustness of Shadow-FT.

- This work highlights the potential of leveraging BASE models to enhance their INSTRUCT counterparts, and we hope it inspires further research and broader applications in the future.

## 2 PRELIMINARIES AND MOTIVATION

### 2.1 BACKGROUND

**Basic tuning methods.** Supervised Fine-tuning (SFT) is a fundamental approach to updating the knowledge of LLMs. Vanilla SFT methods update all the parameters by gradient descent following $W^+ \leftarrow W + \Delta W$, where $W \in \mathbb{R}^{d_1 \times d_2}$ is an arbitrary weight and $W^+$ is the updated variant. To reduce the update costs, LoRA (Hu et al., 2022) introduces a low-rank branch to learn the weight updates following $W^+ \leftarrow W + AB$, where $A \in \mathbb{R}^{d_1 \times r}, B \in \mathbb{R}^{r \times d_2}$ and $r \ll \min\{d_1, d_2\}$. The original weight $W$ is frozen during training, and only the low-rank branch is updated.

**BASE and INSTRUCT.** Current LLMs typically follow a two-stage training pipeline, including pre-training and post-training. During pre-training, LLMs are trained on massive training data on next token prediction tasks (Brown et al., 2020), and the weights would be released as BASE version. The INSTRUCT variant, post-trained upon the BASE model, is further aligned with human preference and tuned for reasoning tasks (Ouyang et al., 2022). Therefore, INSTRUCT model performs better than BASE model regarding instruction-following ability.

---

[1]https://huggingface.co/datasets/BAAI/Infinity-Instruct

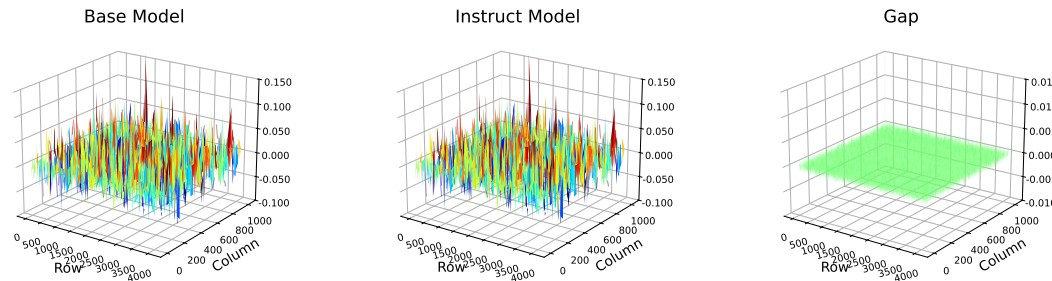

Figure 2: Weight distributions for Llama-3.1-8B. We visualize the same linear layer (layer.0.k_proj) for BASE model (left), INSTRUCT model (middle), and their gap (right). Though zoomed in 10x in the z-axis, the gap is negligible and the average $\sigma$ value is 0.016.

## 2.2 DIRECTLY TUNING INSTRUCT

However, tuning the INSTRUCT models often leads to marginal improvements and even performance degeneration. Table 1 shows the tuned performance of the INSTRUCT models using the BAAI-2k. We report the average scores of popular benchmarks. Compared to the vanilla INSTRUCT, the tuned version shows marginal improvement, and even degeneration in more cases. Specifically, as shown in Table 1, tuning Qwen-3-4B on the BAAI-2k dataset via conventional LoRA would lead to a drop of 2.6 in Math-7 (from 73.8 to 71.2), 6.8 in Code-3 (from 66.4 to 59.6), and 2.6 in Knowledge-9 (from 63.7 to 61.1). Therefore, how to effectively tune INSTRUCT remains a challenge.

## 2.3 SIMILAR WEIGHTS: BASE & INSTRUCT

Fortunately, we observe that the weights of BASE and INSTRUCT are highly similar. To calculate the similarity, we first define the relative gap ratio $\sigma$ as follows:

$$\sigma = \frac{\sum |W_B - W_I|}{\sum |W_B| + \sum |W_I|}, \tag{1}$$

where $\sum$ is the element-wise sum and $|\cdot|$ means the absolute operations. The $\sigma$ would be 1 if one is much larger than the other, and be 0 if the two matrices are exactly the same. The smaller the $\sigma$, the more similar the two matrices are. Figure 2 shows the weights of the same layer from BASE and INSTRUCT, and also their differences with $\sigma = 0.016$. We can find that the gaps are quite small and negligible after zooming in 10x in the z-axis. Please refer to Appendix C for more $\sigma$ regarding various LLMs. In summary, these paired BASE and INSTRUCT models are highly similar with $\sigma < 0.03$.

## 3 METHODOLOGY

### 3.1 SHADOW-FT

To tackle the issue that directly tuning INSTRUCT fails, we propose a novel framework, Shadow-FT, to tune the INSTRUCT on BASE. Motivated by the observation that BASE and INSTRUCT models are highly similar, we argue that the weight updates of BASE can be directly added to INSTRUCT. Since they share the same structures, no extra operations are required. Specifically, in Shadow-FT, we first tune the BASE model:

$$W_B^+ \leftarrow Tune(W_B), \tag{2}$$

where $Tune$ is the fine-tuning method, such as full-parameter fine-tuning and LoRA. After that, we would like to get the weights updates as the learned knowledge, and directly graft these updates to the INSTRUCT model as:

$$W_I^+ = W_I + (W_B^+ - W_B) = W_I + (Tune(W_B) - W_B). \tag{3}$$

Traditional tuning on INSTRUCT can be formulated as:

$$W_I^+ = W_I + (W_I^+ - W_I) = W_I + (Tune(W_I) - W_I). \tag{4}$$

We can find that Shadow-FT introduces no extra training costs. The only difference is the basic weights to learn the weight updates for INSTRUCT model. Vanilla FT methods rely on the INSTRUCT model while Shadow-FT on the BASE model. Since the BASE version is pre-trained only, we believe that the weight updates would be more suitable for modeling the knowledge with less priority, compared to updates of the INSTRUCT version.

## 3.2 RELATION WITH TASK VECTORS

Task Vectors aim to represent the ability on tasks as vectors, and are widely used for arithmetic operations on these tasks regarding the same base model (Ilharco et al., 2022). Chat Vector (Huang et al., 2023) extends such an idea to LLMs, which models weight differences between INSTRUCT and BASE models as vectors and then adds the vectors to continually pretrained BASE models. Specifically, Chat Vector continually pre-trains Llama2 (Touvron et al., 2023) on the Traditional Chinese corpus, and then adds on the chat vectors. Compared to Chat Vector (Huang et al., 2023), the differences are as follows: 1) *task*: Chat Vector focuses on continual pertaining while Shadow-FT can be applied to board tuning methods, including full-parameter fine-tuning, LoRA, and DPO. 2) *motivation*: Chat Vector aims to extend the language ability. Shadow-FT aims to tackle the degeneration issue based on the weight similarity.

## 4 EXPERIMENTS

### 4.1 EXPERIMENTAL SETUP

**Training.** For the tuning data, we build BAAI-2k by extracting 2000 samples from BAAI-Infinity-Instruct Dataset [2] following (Zhou et al., 2023; Muennighoff et al., 2025). We select the samples with high rewards to ensure the data quality and uniform sampling among all categories for data diversity. Without loss of generality, we tune various LLMs, including Qwen 3 series (Team, 2025) and Llama 3 series (AI@Meta, 2024). Also, we report the results on Gemma-3 series (Team et al., 2025a), Yi series (Young et al., 2024), and Falcon series (Almazrouei et al., 2023) in Section 5.7. We employ LLaMA-Factory (Zheng et al., 2024) for the code base and apply two tuning strategies: full-parameter and LoRA. All experiments are conducted on 8 A100 GPUs. Please refer to Appendix D.1 for detailed hyperparameters.

**Evaluation.** To evaluate the tuned LLMs on downstream benchmarks, we employ the OpenCompass framework (Contributors, 2023b) and lmdeploy as the acceleration framework (Contributors, 2023a). During inference, we set the cutoff length as 4096 and the batch size as 512. Considering the benchmarks, we select three representative abilities, i.e., mathematical, coding, and commonsense reasoning ability, and report the average scores marked as Math-7, Code-3, and Knowledge-9. Specifically, Math-7 denotes the results of AIME24 MAA (2024), GSM8K (0-shot and 8-shot) Cobbe et al. (2021), MATH Hendrycks et al. (2021b), MATH-500, Minerva_Math Lewkowycz et al. (2022), SVAMP Patel et al. (2021). Code-3 for HumanEval Chen et al. (2021a), HumanEval+ Liu et al. (2023), LiveCodeBench Jain et al. (2024). Knowledge-9 for ARC-challenge Clark et al. (2018), BBH (0-shot and few-shot), DROP Dua et al. (2019), GPQA Diamond Rein et al. (2024), MMLU Hendrycks et al. (2021a), MMLU Pro Wang et al. (2024b), Winogrande ai2 (2019), TheoremQA Chen et al. (2023). To avoid the impact of different prompts, we mainly evaluate under a zero-shot setting. Please refer to Appendix A for more details. For Qwen-3 series, we adapt enable_thinking as false for universal evaluations, and we report pass@k results of both thinking and non-thinking in Appendix D.4.

### 4.2 MAIN RESULTS

Table 1 shows the results of tuning various mainstream LLMs on BAAI-2k using full-parameter fine-tuning and LoRA. We set the rank as 128 in LoRA. Some findings can be summarized as follows:

- **Conventional tuning methods lead to marginal improvements and even performance degeneration.** Considering the average score, we can find that conventional tuning methods bring marginal improvements, such as 74.8 vs. 74.5 on Qwen-2.5-32B and 47.4 vs. 47.5

---

[2]https://huggingface.co/datasets/BAAI/Infinity-Instruct/tree/main/Gen

**Table 1:** Performance comparison of different methods tuning popular LLMs. **Math-7** denotes the average score of 7 mathematical benchmarks including AIME24, **Code-3** for 3 code benchmarks including LiveCodeBench, and **Knowledge-9** for 9 commonsense reasoning benchmarks including MMLU Pro. For **Math-7** and **Code-3**, we report the mean value of three runs. We employ the Instruct version and report the final average scores. Please refer to Appendix D.2 for detailed scores.

| Model | Method | Math-7 | | Code-3 | | Knowledge-9 | | Avg. |
|---|---|---|---|---|---|---|---|---|
| | | *Full* | *LoRA* | *Full* | *LoRA* | *Full* | *LoRA* | |
| Qwen-3-4B | Instruct | 73.8 | | 66.4 | | 63.7 | | 68.0 |
| | FT | 72.9 | 71.2 | 66.4 | 59.6 | 62.9 | 61.1 | 65.7 |
| | Shadow-FT | 73.7 | 75.9 | 67.4 | 69.7 | 64.9 | 65.0 | **69.4** |
| Qwen-3-8B | Instruct | 74.5 | | 72.7 | | 64.7 | | 70.6 |
| | FT | 74.0 | 71.3 | 71.2 | 69.6 | 64.6 | 64.3 | 69.2 |
| | Shadow-FT | 75.9 | 74.8 | 73.1 | 71.9 | 65.6 | 67.8 | **71.5** |
| Qwen-3-14B | Instruct | 75.8 | | 76.8 | | 71.2 | | 74.6 |
| | FT | 75.2 | 73.3 | 76.2 | 74.4 | 70.6 | 70.4 | 73.4 |
| | Shadow-FT | 78.9 | 78.6 | 77.0 | 77.8 | 71.4 | 71.5 | **75.9** |
| Qwen-2.5-32B | Instruct | 74.1 | | 75.9 | | 73.4 | | 74.5 |
| | FT | 75.7 | 74.3 | 75.8 | 75.9 | 73.6 | 73.8 | 74.8 |
| | Shadow-FT | 74.9 | 75.7 | 76.1 | 76.2 | 73.5 | 73.8 | **75.0** |
| Llama-3.2-1B | Instruct | 23.8 | | 26.5 | | 34.2 | | 28.2 |
| | FT | 24.5 | 25.3 | 26.1 | 26.6 | 32.8 | 33.3 | 28.1 |
| | Shadow-FT | 25.2 | 27.2 | 28.2 | 27.9 | 32.7 | 32.3 | **29.0** |
| Llama-3.2-3B | Instruct | 53.6 | | 39.3 | | 49.3 | | 47.4 |
| | FT | 52.7 | 51.9 | 40.2 | 41.4 | 49.4 | 49.1 | 47.5 |
| | Shadow-FT | 54.9 | 56.2 | 40.3 | 42.8 | 49.5 | 48.9 | **48.8** |
| Llama-3.1-8B | Instruct | 56.8 | | 50.9 | | 56.6 | | 54.8 |
| | FT | 56.8 | 57.8 | 53.4 | 51.8 | 58.5 | 57.5 | 56.0 |
| | Shadow-FT | 58.7 | 59.4 | 51.8 | 50.9 | 57.6 | 58.7 | **56.2** |

on Llama-3.2-3B. Moreover, they would lead to performance degeneration, such as 68.0 vs. 65.7 on Qwen-3-4B and 70.6 vs. 69.2 on Qwen-3-8B. The observations are consistent across full-parameter tuning and LoRA.

- **While conventional tuning fails, Shadow-FT performs well in adaptation at the same cost.** Across all model sizes and tasks, Shadow-FT consistently outperforms tuning baselines and vanilla INSTRUCT model. For example, on Qwen-3-4B, Shadow-FT archives an average score of 69.4, which is 3.7 higher than the 65.7 of conventional tuning methods and 1.4 higher than the vanilla INSTRUCT model. The conclusion is consistent on larger models such as Qwen-3-14B. Moreover, Shadow-FT does not introduce any extra training overheads. These consistent gains demonstrate that our proposed Shadow-FT can effectively learn the knowledge contained in training data.

- **Shadow-FT works well under both full-parameter setting and LoRA.** For instance, when tuning Qwen-3-4B under full-parameter setting, Shadow-FT achieves 73.7/67.4/64.9 on Math-7/Code-3/Knowledge-9 compared to 72.9/66.4/62.9 of conventional tuning methods. When applying a low-rank setting, Shadow-FT achieves 75.9/69.7/65.0, which is 4.7/10.1/3.9 higher than conventional LoRA. These indicate that Shadow-FT is effective with different tuning strategies, showing its robustness.

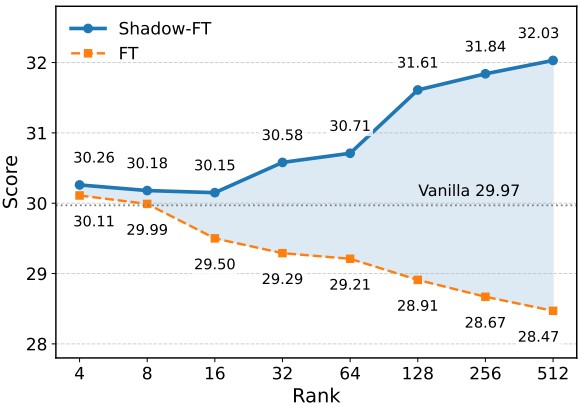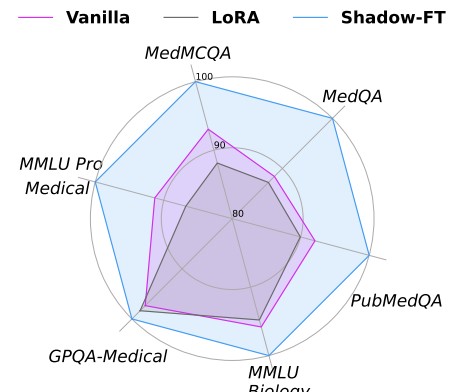

Figure 3: The average of Math-7, Code-3, and Knowledge-9 for different ranks when tuning Llama-3.2-1B using LoRA. We report the best performance searching learning rates in {5e-5, 1e-4, 2e-4, 5e-4}.

Figure 4: Performance of various methods when tuning Llama-3.2-1B on the Medical-o1-reasoning-SFT dataset. Detailed scores at Table 7.

- **LoRA can outperform full-parameter.** When tuning using our BAAI-2k dataset, we find that Shadow-FT (LoRA) can outperform Shadow-FT (full), such as 69.7 vs. 67.4 on Code-3 when tuning Qwen-3-4B. Interestingly, we find that Shadow-FT (LoRA) typically performs better than Shadow-FT (full) on Math-7. However, considering the conventional tuning methods, FT (full) would perform better (Biderman et al., 2024), such as 75.9 vs. 74.8 on Qwen-3-8B. We leave it to future work for further investigation.

Moreover, we further conduct case study on Llama-3.1-8B-Instruct. Please refer to Appendix E for more details.

## 5 EXTENSIVE ANALYSIS

### 5.1 RANKS IN LORA

We fine-tune the Llama-3.2-1B using LoRA with different ranks (from 4 to 512), and report the average scores after searching learning rates in {5e-5, 1e-4, 2e-4, 5e-4}. As shown in Figure 3, our proposed Shadow-FT (LoRA) can always outperform conventional LoRA with different ranks, demonstrating the robustness. With a larger rank, the conventional LoRA would perform worse, indicating more severe degeneration when tuning the INSTRUCT model (Yang et al., 2024a). In contrast to that, Shadow-FT (LoRA) can consistently benefit from more parameters (with larger ranks) and achieves better performance. For the results on Llama-3.1-8B, please refer to Appendix D.5.

### 5.2 TUNING ON DOMAIN DATA

Tuning methods are typically employed to adapt LLMs for a specific domain, such as medical. Therefore, we perform tuning experiments on specific domain data, including Medical-o1-reasoning-SFT (Chen et al., 2024) in the medical domain, Code-Z1 (Yu et al., 2025) in the code domain, and LIMO Ye et al. (2025) & OpenR1-Math (Face, 2025) in the math domain. Following LIMO Ye et al. (2025), we uniformly down sample the Medical-o1-reasoning-SFT to 1,000, and Code-Z1/OpenR1-Math to 2,000. On these domain tasks, we employ the LoRA with rank 128 and optimize with a learning rate of 2e-4.

Figure 4 reports the results of tuning Llama-3.2-1B on Medical-o1-reasoning-SFT. We report the results on MMLU Pro-Medical (Wang et al., 2024b), MedMCQA (Pal et al., 2022), PubMedQA (Jin et al., 2019), MMLU-Biology (Yue et al., 2024), and GPQA-Medical following (Chen et al., 2024), while normalizing the maximum score to 1 for better visualization. We can find that conventional LoRA would lead to performance degeneration, while Shadow-FT (LoRA) improves the performance,

**Table 2:** The detailed mathematical and code performance tuning Qwen-3-8B and Llama-3.1-8B on Code-Z1, LIMO, and OpenR1-Math. *Ins.* denotes the vanilla INSTRUCT baseline, *LoRA* for conventional LoRA, and *Shadow* for proposed Shadow-FT (LoRA). Green↓/Red↑ indicates a performance drop/gain relative to the vanilla INSTRUCT baseline.

| Benchmark | Qwen-3-8B | | | | | | | Llama-3.1-8B | | | | | | |
| | | Code-Z1 | | LIMO | | OpenR1-Math | | | Code-Z1 | | LIMO | | OpenR1-Math | |
| | *Ins.* | *LoRA* | *Shadow* | *LoRA* | *Shadow* | *LoRA* | *Shadow* | *Ins.* | *LoRA* | *Shadow* | *LoRA* | *Shadow* | *LoRA* | *Shadow* |
|---|---|---|---|---|---|---|---|---|---|---|---|---|---|---|
| AIME24 | 20.0 | 13.3 | 36.7 | 23.3 | 26.7 | 16.7 | 26.7 | 6.7 | 3.3 | 20.0 | 6.7 | 3.3 | 3.3 | 6.7 |
| GSM8K(8shot) | 87.4 | 84.1 | 88.3 | 85.2 | 88.7 | 83.1 | 86.8 | 84.2 | 84.1 | 85.8 | 80.5 | 83.8 | 82.3 | 84.8 |
| GSM8K(0shot) | 93.0 | 91.9 | 93.6 | 91.7 | 92.4 | 92.7 | 92.9 | 84.2 | 85.4 | 85.7 | 82.5 | 86.1 | 86.1 | 85.9 |
| MATH | 70.9 | 69.4 | 69.1 | 70.0 | 67.6 | 70.6 | 66.5 | 48.0 | 48.8 | 51.3 | 44.3 | 45.8 | 39.8 | 47.7 |
| MATH-500 | 83.2 | 79.8 | 88.0 | 77.0 | 80.4 | 80.2 | 85.0 | 48.4 | 50.8 | 55.4 | 44.4 | 43.8 | 41.8 | 48.8 |
| Minerva_Math | 73.0 | 69.7 | 72.9 | 69.9 | 73.1 | 70.8 | 73.2 | 40.6 | 39.6 | 45.5 | 37.1 | 41.2 | 44.0 | 44.2 |
| SVAMP | 91.4 | 90.3 | 93.3 | 90.9 | 92.9 | 90.3 | 93.0 | 83.1 | 86.5 | 86.9 | 83.7 | 85.9 | 85.1 | 87.1 |
| **Math-7** | 74.5◇ | 71.2↓ | **77.4**↑ | 72.6↓ | **75.1**↑ | 72.1↓ | **75.7**↑ | 56.8◇ | 57.1↑ | **61.5**↑ | 54.2↓ | **55.7**↓ | 54.6↓ | **58.0**↑ |
| HumanEval | 84.2 | 82.3 | 87.8 | 84.2 | 86.0 | 78.1 | 83.5 | 71.3 | 64.6 | 70.1 | 68.9 | 70.7 | 72.6 | 72.0 |
| HumanEval+ | 79.9 | 76.8 | 78.1 | 79.3 | 81.1 | 75.6 | 81.1 | 63.4 | 48.2 | 64.6 | 62.2 | 64.0 | 61.6 | 62.8 |
| LiveCodeBench | 51.5 | 43.2 | 54.7 | 48.7 | 53.1 | 47.6 | 54.6 | 19.8 | 11.8 | 20.5 | 18.6 | 20.7 | 15.6 | 19.9 |
| **Code-3** | 72.7◇ | 67.4↓ | **73.5**↑ | 70.7↓ | **73.4**↑ | 67.1↓ | **73.1**↑ | 50.9◇ | 41.5↓ | **51.7**↑ | 49.9↓ | **51.8**↑ | 49.9↓ | **51.6**↑ |

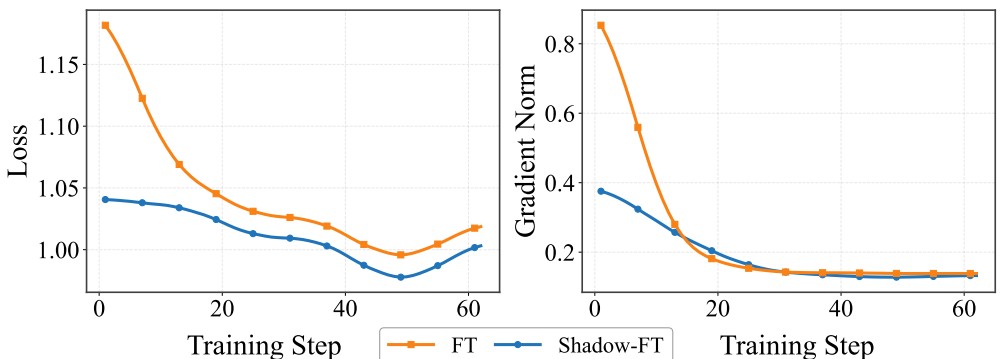

Figure 5: Optimization dynamics on loss and gradient when tuning Qwen3-8B via INSTRUCT (i.e., FT) and BASE (i.e., Shadow-FT).

which is consistent with the conclusion on BAAI-2k. Besides, we further report the results directly tuning the BASE. Please refer to Appendix D.3 for detailed scores.

Table 2 shows the detailed results of Math-7 and Code-3 tuning Qwen-3-8B and Llama-3.1-8B on Code-Z1, LIMO, and OpenR1-Math. The observations are consistent, i.e., conventional LoRA would lead to degeneration, while the proposed shadow-FT (LoRA) can effectively adapt LLMs on specific domain knowledge. For instance, Shadow-FT (LoRA) achieves a Math-7 score of 77.4 on Qwen-3-8B, which is 6.2 higher than 71.2 of LoRA, and 2.9 higher than the vanilla INSTRUCT model. Moreover, we also find that tuning LLM via Shadow-FT on code data can improve the math capability (Yu et al., 2025), and vice versa. In particular, when tuned via shadow-FT on Code-z1, the Qwen-3-8B can achieve a score of 36.7 on the tough AIME-24 benchmark, showing superior adaptation and generalization ability.

### 5.3 MECHANISTIC ANALYSIS OF OPTIMIZATION DYNAMICS

To provide insight into why Shadow-FT outperforms vanilla FT, we further analyze the optimization dynamics of both methods from a loss and gradient perspective. We denote the loss and gradient for the INSTRUCT model (tuned with vanilla FT) as **Loss(I)** and **Grad(I)**, and for the BASE model (tuned with Shadow-FT) as **Loss(S)** and **Grad(S)**.

**Table 3:** Performance of tuning Llama-3.1-8B using DPO and Shadow-DPO with ranks be 8 and 128, respectively.

| Method | Rank | 19-Avg. |
|---|---|---|
| Vanilla | – | 54.77 |
| DPO | 8 | 54.80 |
| Shadow-DPO | 8 | **54.96** |
| DPO | 128 | 54.62 |
| Shadow-DPO | 128 | **55.39** |

**Table 4:** Performance of Gemma-3 and Llama-3.2-Vision on the multi-modal ChartQA task. We set the rank of LoRA to 128.

| Model | Size | Vanilla | LoRA | Shadow-FT |
|---|---|---|---|---|
| Gemma-3 | 12B | 37.36 | 53.48 | **54.92** |
| | 27B | 41.92 | 60.28 | **63.80** |
| Llama-3.2-Vision | 11B | 22.12 | **74.44** | 74.12 |
| | 90B | 30.92 | 79.92 | **80.60** |

Figure 5 illustrates these metrics while tuning Qwen3-8B with LoRA (r=128). The two approaches show markedly different training dynamics. *At initialization*, **Loss(I)** is 22.6% higher than **Loss(S)**, and **Grad(I)** is 3.25× larger, reflecting a poor task fit and strong resistance from the instruction-following prior (Ji et al., 2024). *During training*, **Grad(I)** decays precipitously (11/61 step), while **Grad(S)** decreases more moderately (58%), indicating that INSTRUCT quickly enters a rigid optimization regime with suppressed updates while Shadow-FT sustains enjoying a smoothed learning. *By convergence* (step 61/61), both gradients stabilize at similar magnitudes, but **Loss(I)** remains 2.4% higher, evidencing inferior adaptation.

Overall, the figure reveals a fundamental contrast in optimization. Vanilla FT on the INSTRUCT model is hindered by a large initial gradient that rapidly diminishes, while the BASE model can avoid this and shows a stable trajectory. However, BASE model is a good learner but a poor backbone due to the lack of post-training. The conclusion is consistent with Pass@k results detailed in Appendix D.4. Therefore, we propose Shadow-FT to learn on BASE and execute on INSTRUCT.

## 5.4 COMBINED WITH DPO

Direct Preference Optimization (DPO), which directly optimizes a language model to adhere to human preferences without explicit reward modeling or reinforcement learning, shows promising performance when applying RL to LLMs (Rafailov et al., 2023). Therefore, we try to combine Shadow-FT with DPO, i.e., applying DPO on BASE and then grafting the weight to INSTRUCT, termed as Shadow-DPO. Specifically, we achieve Shadow-DPO using LoRA on 1,000 paired samples from the Math-Step (Lai et al., 2024) dataset and set the rank to 8 and 128. As shown in Table 3, shadow-DPO outperforms DPO under two settings, such as 55.39 vs. 54.62 of vanilla DPO. It shows that the strategy employing the BASE as proxy of INSTRUCT also works for DPO. Meanwhile, a larger rank leads to better results for shadow-DPO, which is consistent with results tuning on BAAI-2k shown in Figure 3.

## 5.5 PERFORMANCE ON MLLM

For generality, we further conduct experiments tuning Multimodal Large Language Models (MLLMs). For the dataset, we select 10,000 samples from ChartMoE (Xu et al., 2024), which takes a chart and a natural language question as input to predict the answer. For MLLM, we select Gemma-3 (Team et al., 2025a) 12B/27B and Llama-3.2-Vision (Grattafiori et al., 2024) 11B/90B. During training, we employ LoRA and set the rank to 128. The learning rate is 2e-4. We evaluate the tuned model via lmms-eval framework (Zhang et al., 2024). As shown in Table 4, both conventional LoRA and Shadow-FT (LoRA) effectively adapt MLLMs on ChartQA (Masry et al., 2022) tasks. Meanwhile, our proposed Shadow-FT outperforms LoRA, especially on larger models, such as 63.80 on Gemma-3-27B compared to 60.28 of vanilla LoRA and 80.6 on Llama-3.2-Vision-90B compared to 79.92.

## 5.6 WEIGHT DELTA SCALING

In Shadow-FT, we directly graft the learned weights from BASE to INSTRUCT. We further explore the scaling of learned weights. Please refer to Appendix D.6 for more details. In summary, our proposed Shadow-FT outperforms vanilla INSTRUCT with different scaling factors, showing strong

robustness. Moreover, a factor slightly larger than 1 would yield better results, while we leave it for future work to explore the best factor.

## 5.7 MODEL ZOO: MORE LLMS

We further apply Shadow-FT to more LLMs, including Gemma-3 series (Team et al., 2025a), Yi series (Young et al., 2024), and Falcon series (Almazrouei et al., 2023). Please refer to Appendix B for more details. We can find that proposed Shadow-FT consistently outperforms conventional tuning methods. All the tuned models will be made public in the future.

## 6 RELATED WORK

### 6.1 TUNING FOR LLMS

Large language models (LLMs) gain superior ability from pre-training on tremendous data (Gururangan et al., 2020), followed by tuning on various downstream tasks (Ouyang et al., 2022; Muennighoff et al., 2025). These methods can be categorized into: 1) full-parameters method, which updates all the parameters, and 2) parameter-efficient fine-tuning (PEFT) method, lowering the tuning costs via parameter selection (Zaken et al., 2021) or low-rank branches (Hu et al., 2022; Wu et al., 2024b). More recently, Reinforcement Learning from Human Feedback (RLHF) methods show promising performance in aligning models to human preferences and improving the reasoning ability Rafailov et al. (2023); Bai et al. (2023); Guo et al. (2025); Team et al. (2025b). These methods focus on improving the training strategy and involve the target model only. In this paper, we propose Shadow-FT to tune INSTRUCT model on BASE model. Also, our proposed Shadow-FT can be combined with these baselines to enhance the performance.

### 6.2 MODEL GUIDANCE IN TUNING

Introducing extra knowledge from other models has been proven as a promising way to enhance tuning performance, such as knowledge distillation (Hinton et al., 2015; Wu et al., 2024a) and proxy-tuning (Liu et al., 2024). Knowledge distillation methods aim to transfer the knowledge from a larger teacher model to a compact student model, via aligning the outputs (Wu et al., 2024a; Yang et al., 2024b) or employing the teacher's outputs as training data (Qin et al., 2024; Min et al., 2024). Proxy-tuning first tunes a smaller LLM and then applies the logit differences to a larger model (Liu et al., 2024). These methods transfer knowledge at the feature level or data level, while our proposed Shadow-FT directly grafts the weight updates. RE-Adapt (Fleshman & Durme, 2024) also utilizes the Base/Instruct model pair for adaptation. However, RE-Adapt models the static weight difference with a *low-rank* approximation, whereas Shadow-FT is a model-free approach that directly transfers the full dynamic updates without any assumption. Additionally, we notice a very recent concurrent work (Lin et al., 2025) to transfer the fine-tuning ability. Differently, our proposed Shadow-FT focuses on tuning INSTRUCT via BASE model based on the *observation* that the weights are highly similar. Moreover, we conduct experiments on more LLMs across more benchmarks, and further extend the idea to MLLMs and DPO.

## 7 CONCLUSION

In this work, we propose Shadow-FT, a novel framework to fine-tune INSTRUCT models by leveraging their corresponding BASE models. Inspired by the observation that the weights of BASE and INSTRUCT are highly similar, we propose Shadow-FT to tune INSTRUCT vis BASE, aiming to tune INSTRUCT better. Extensive experiments across multiple LLM series, including Qwen, Llama, Gemma, and Falcon, demonstrate that Shadow-FT consistently outperforms conventional full-parameter and parameter-efficient fine-tuning methods. Notably, Shadow-FT introduces no additional training cost or parameters, yet it achieves superior performance across diverse benchmarks covering math, coding, and reasoning tasks. We further show that Shadow-FT generalizes well to multimodal large language models (MLLMs) and can be seamlessly combined with alignment techniques such as DPO, offering a simple yet effective solution for improving instruction-following models.

## REPRODUCIBILITY STATEMENT

We are committed to the reproducibility of our work. The MATH-7 results, averaged across three trials, are shown in Table 1 and Table 8. The full source code required to reproduce our experiments is included in the supplementary material. Corresponding hyperparameters and detailed configuration files for all experiments are documented in Section 4.1. All experiments were conducted on publicly available benchmarks, and the details are provided in Appendix A.

## ETHICS STATEMENT

We have adhered to the ICLR Code of Ethics in this research. Our work is based entirely on publicly available models and benchmarks, involves no human subjects, and we commit to releasing our code to ensure reproducibility.

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

# LLM USAGE DISCLOSURE

The human authors are primarily responsible for this work. We utilized several large language models (e.g., GPT-4, Gemini Pro, Claude 3) as general-purpose assistive tools to improve the quality of our research and writing. Their use was limited to the following specific tasks: assisting with code implementation and debugging, generating boilerplate code, refining the language and formatting of the manuscript, and proofreading. The authors conceived all research ideas, designed the experiments, and performed the final analysis of the results. We take full responsibility for all content in this paper and confirm that it complies with relevant licenses and ethical guidelines.

# A  BENCHMARKS DETAILS

**Table 5:** Details on instruction-model evaluations. CoT denotes the chain-of-thought setting.

| Evaluation | Metric | Type | n-shot | CoT |
|---|---|---|---|---|
| *Math-7* | | | | |
| **AIME24** | pass@1 | sampling | 0-shot | |
| **GSM8K(0-shot)** | Accuracy | sampling | 0-shot | ✓ |
| **GSM8K(8-shot)** | Accuracy | sampling | 8-shot | ✓ |
| **MATH** | Accuracy | sampling | 0-shot | ✓ |
| **MATH-500** | Accuracy | sampling | 0-shot | |
| **Minerva Math** | Accuracy | sampling | 4-shot | |
| **SVAMP** | Accuracy | sampling | 0-shot | |
| *Code-3* | | | | |
| **HumanEval** | pass@1 | sampling | 0-shot | |
| **HumanEval+** | pass@1 | sampling | 0-shot | |
| **LiveCodeBench** | | average | | |
| - generation | pass@1 | sampling | 0-shot | ✓ |
| - test | pass@1 | sampling | 0-shot | ✓ |
| - prediction | pass@1 | sampling | 0-shot | ✓ |
| *Knowledge-9* | | | | |
| **ARC-Challenge** | Accuracy | sampling | 0-shot | ✓ |
| **BBH(0-shot)** | Accuracy | sampling | 0-shot | |
| **BBH(3-shot)** | Accuracy | sampling | 3-shot | |
| **Drop** | Accuracy | sampling | 0-shot | |
| **GPQA Diamond** | Accuracy | sampling | 0-shot | ✓ |
| **MMLU** | Accuracy | sampling | 0-shot | |
| **MMLU Pro** | Accuracy | sampling | 0-shot | |
| **Winogrande** | Accuracy | sampling | 0-shot | |
| **TheoremQA** | Accuracy | sampling | 0-shot | |

The details about the benchmarks are detailed in Table 5. Since the n-shot setting are unstable, we prefer to report the 0-shot results. For the popular GSM8K and BBH, we also report 8-shot and 3-shot results.

# B  MODEL ZOO: MORE LLMS

We further apply Shadow-FT to more LLMs, including Gemma-3 series (Team et al., 2025a), Yi series (Young et al., 2024), and Falcon series (Almazrouei et al., 2023). The hyperparameters are the same as tuning Qwen 3 and Llama 3. Table 6 shows the results of Math-7, Code-3, and Knowledge-9. We can find that proposed Shadow-FT consistently outperforms conventional tuning methods. For instance, Shadow-FT gets an average of 52.55 when tuning Gemma-3-4B, which is 1.1 higher than the vanilla INSTRUCT model and 7.91 higher than conventional tuning methods.

**Table 6:** Performance comparison of different methods tuning more LLMs. We employ the Instruct version and report the final average scores.

| Model | Method | Math-7 Full | Math-7 LoRA | Code-3 Full | Code-3 LoRA | Knowledge-9 Full | Knowledge-9 LoRA | Avg. |
|---|---|---|---|---|---|---|---|---|
| *Falcon Family* | | | | | | | | |
| Falcon3-3B | Vanilla | 53.33 | | 38.09 | | 47.28 | | 46.23 |
| | FT | 55.83 | 58.70 | 39.57 | 41.23 | 48.43 | 49.50 | 48.88 |
| | Shadow-FT | 56.74 | 60.31 | 41.02 | 43.69 | 48.16 | 48.25 | **49.70** |
| Falcon3-10B | Vanilla | 57.23 | | 60.03 | | 53.85 | | 57.04 |
| | FT | 59.33 | 68.74 | 60.95 | 61.54 | 54.17 | 55.72 | **60.08** |
| | Shadow-FT | 58.27 | 70.40 | 61.35 | 62.20 | 53.19 | 52.83 | 59.71 |
| *Gemma Family* | | | | | | | | |
| Gemma-3-4B | Vanilla | 54.02 | | 48.33 | | 52.01 | | 51.45 |
| | FT | 35.34 | 49.12 | 48.15 | 43.03 | 43.83 | 48.37 | 44.64 |
| | Shadow-FT | 56.68 | 56.30 | 48.87 | 48.93 | 52.88 | 51.62 | **52.55** |
| Gemma-3-12B | Vanilla | 60.82 | | 58.06 | | 61.54 | | 60.14 |
| | FT | 56.56 | 62.84 | 58.17 | 59.21 | 61.63 | 61.99 | 60.07 |
| | Shadow-FT | 61.05 | 64.59 | 58.17 | 60.86 | 61.59 | 62.66 | **61.49** |
| *Yi Family* | | | | | | | | |
| Yi-6B | Vanilla | 17.34 | | 8.40 | | 38.63 | | 21.46 |
| | FT | 18.93 | 18.39 | 10.64 | 11.89 | 40.84 | 40.46 | **23.53** |
| | Shadow-FT | 17.73 | 17.21 | 13.35 | 14.30 | 38.70 | 38.25 | 23.26 |
| Yi-Coder-9B | Vanilla | 28.01 | | 61.85 | | 40.73 | | 43.53 |
| | FT | 26.05 | 26.11 | 52.70 | 53.95 | 39.55 | 37.22 | 39.26 |
| | Shadow-FT | 28.41 | 29.09 | 62.07 | 64.72 | 40.27 | 39.88 | **44.07** |

## C  SIMILARITY ON MORE LLMS

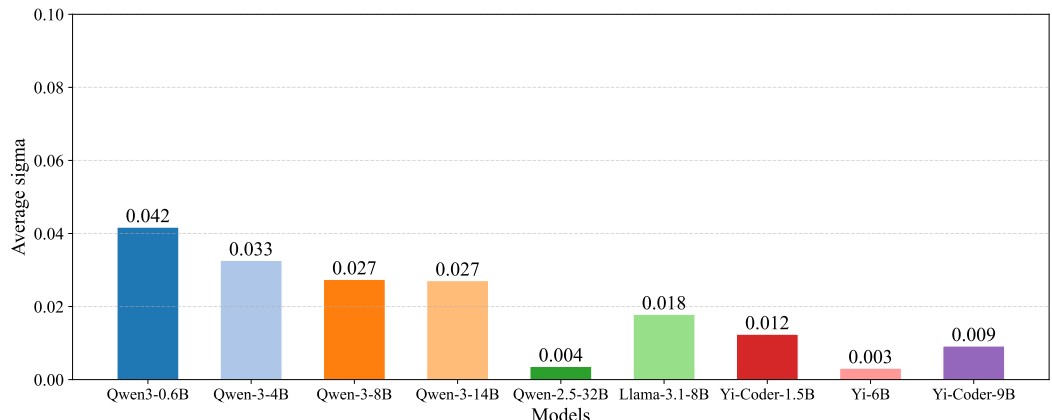

Figure 6: Average $\sigma$ values of more LLMs.

As shown in Figure 6, we can find that all $\sigma < 0.05$, indicating high similarity between BASE and INSTRUCT. Also, the larger the LLMs, the smaller the gaps.

# D EXPERIMENTAL DETAILS

## D.1 HYPER-PARAMETERS

For the experiments, we set the hyperparameters after grid search. The batch size is 2, with the gradient_accumulation_steps as 16. During experiments, the cutoff the inputs to 4096 and train for 1 epoch.

## D.2 DETAILED DATA OF TABLE 1

The detailed scores are listed in Table 9, Table 10, and Table 11.

## D.3 DETAILED TABLE FOR MEDICAL BENCHMARKS

Table 7 reports the detailed results of tuning Llama-3.2-1B on the Medical-o1-reasoning-SFT dataset. Shadow-FTR donates the method integrating fine-tuned weights from INSTRUCT to BASE.

**Table 7:** Performance of LLAMA-3.2-1B-INSTRUCT on medical QA benchmarks.

| Benchmark | Tune on Instruct | | | Tune on Base | | |
|---|---|---|---|---|---|---|
| | Instruct | FT | Shadow-FT | Base | Base-FT | Shadow-FTR |
| GPQA-Medical | 23.85 | 24.10 | 24.50 | **25.25** | 25.00 | 24.75 |
| MMLU Pro-Medical | 25.20 | 23.95 | **27.60** | 13.10 | 12.60 | 12.30 |
| MedMCQA | 30.15 | 28.55 | **32.40** | 30.20 | 30.60 | 29.50 |
| MedQA | 25.95 | 25.60 | 29.35 | 29.80 | **30.35** | 29.45 |
| PubMedQA | 55.85 | 54.55 | **60.65** | 49.35 | 51.90 | 50.20 |
| Avg. | 32.20 | 31.35 | **34.90** | 29.54 | 30.09 | 29.24 |

## D.4 PERFORMANCE ON PASS@K

To evaluate the exploration capability, we use the popular Pass@k, which is defined as the fraction of problems for which at least one correct response is produced in $k$ independent trials. However, directly computing Pass@k using only $k$ rollouts for each problem often suffers from high variance. Therefore, we adapt the unbiased estimator (Chen et al., 2021b). Specifically, we roll out for $n$ times ($n \geq k$), and calculate Pass@k as follows:

$$\text{Pass@}k := \mathbb{E}_{x \sim \mathcal{D}} \left[ 1 - \frac{\binom{n-c}{k}}{\binom{n}{k}} \right], \tag{5}$$

where $x$ is the input prompt from dataset $D$, and $c$ is the count of correct solutions.

We employ the QWEN3-8B model, which supports seamless switching between *Thinking* mode and *Non-thinking* mode. We can easily switch between two modes using one control hyperparameter. We set other hyperparameters following the official report of Qwen3 (Team, 2025): do_sample=True, temperature=0.6, top_k=20, top_p=0.95, max_new_tokens=38912 for bettr alignment.

**Non-thinking mode.** Under non-thinking decoding, absolute PASS@K values are small for all methods, yet SHADOW-FT exhibits a clearer upward trend with larger $k$, progressively surpassing the INSTRUCT baseline. In contrast, vanilla FT yields weak performance and rarely produces correct solutions. Importantly, all methods used the same number of training examples and the same training cutoff length; the only difference is the initialization (BASE vs. INSTRUCT). This comparison suggests that BASE is a better learner for supervised adaptation—its newly acquired knowledge is less prone to collapse.

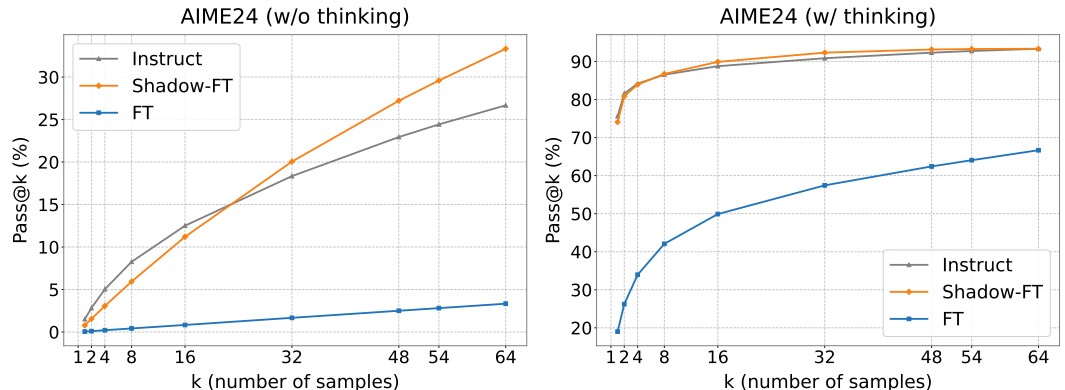

Figure 7: Pass@k performance on Qwen3-8B thinking and non-thinking modes.

**Thinking mode.** Under thinking mode, the effect is more pronounced: SHADOW-FT and INSTRUCT follow similarly steep, rapidly saturating curves that nearly reach the model's capacity limit, whereas vanilla FT maintains a large ($>30\%$) gap across $k$. This pattern implies that vanilla FT hurts the thinking ability of INSTRUCT and makes it less receptive to new knowledge compared with BASE-initialized training.

Across both modes, our proposed Shadow-FT avoids collapse and retains—often enhances the upper-bound competence of the underlying INSTRUCT model. This property is valuable for subsequent RL or other generalization-critical settings (Zhu et al., 2025). We attribute the robustness to the favorable inductive characteristics of BASE-initialized learning, whereas vanilla FT on an INSTRUCT model struggles to achieve the same balance of stability and adaptability.

## D.5 RANKS IN LORA ON LLAMA-3.1-8B

As shown in Figure 8, the conclusions regarding Llama-3.1-8B are consistent with Section 5.1.

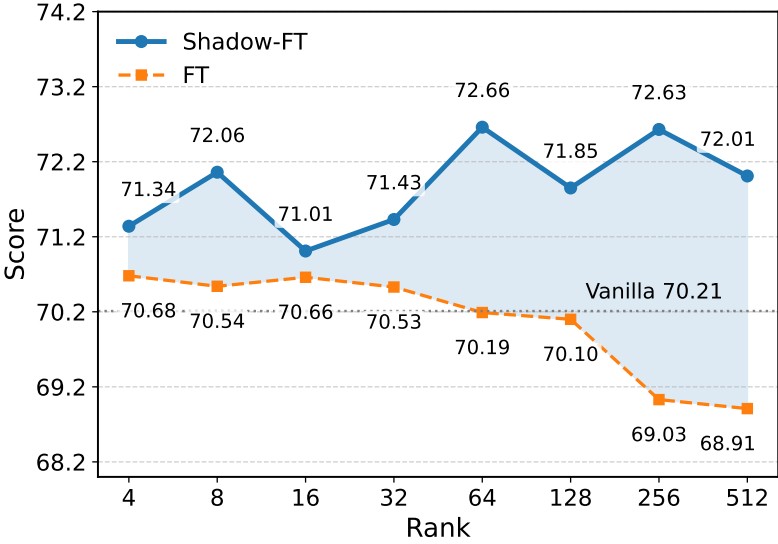

Figure 8: Performance tuning Llama-3.1-8B with different ranks.

## D.6 WEIGHT DELTA SCALING

To further explore the effect of scaling the transferred deltas in the Shadow-FT strategy, we introduce an interpolation design controlled by a scaling factor $\alpha$. Specifically, let $\Delta W_B = W_B^+ - W_B$ denote

the weight updates learned on the BASE model. Instead of directly applying the full deltas to the INSTRUCT model, we interpolate as follows:

$$W_I^+ = W_I + \alpha \cdot \Delta W_B = W_I + \alpha \cdot (W_B^+ - W_B). \qquad (6)$$

Here, $\alpha = 1.0$ recovers the standard Shadow-FT formulation, while $\alpha = 0.0$ reduces to the original INSTRUCT model without transfer. Intermediate values of $\alpha$ provide a smooth interpolation between the two, allowing us to examine how the magnitude of transferred deltas affects downstream performance. The summarized results of Llama-3.1-8B-Instruct on the MATH-7 benchmark are presented in Table 8, showing that $\alpha = 1.0$ yields the excellent overall trade-off, while other ratios offer insights into the sensitivity of tasks to partial transfer. Adaptive scaling strategies (e.g., layer-wise or task-specific factors) are left for future work.

**Table 8:** Detailed results on the math benchmarks (averaged over three repeated runs for each ratio). The ratio of **0.0** denotes **vanilla Instruct**, while **1.0** for the proposed **Shadow-FT**.

| Ratio | Math-7 | AIME24 | GSM8K (8-shot) | GSM8K (0-shot) | MATH | MATH 500 | Minerva Math | SVAMP |
|-------|--------|--------|----------------|----------------|------|----------|--------------|-------|
| 0.0 | 56.2 | 1.1 | 84.8 | 85.0 | 48.5 | 50.7 | 40.5 | 82.7 |
| 0.1 | 56.5 | 1.1 | 85.2 | 84.3 | 49.8 | 51.1 | 41.3 | 83.0 |
| 0.2 | 57.1 | 2.2 | 85.0 | 85.0 | 50.4 | 50.5 | 42.3 | 84.3 |
| 0.3 | 57.9 | 2.2 | 85.0 | 84.4 | 51.3 | 53.5 | 43.8 | 85.0 |
| 0.4 | 58.8 | 6.7 | 85.6 | 83.6 | 52.9 | 54.2 | 44.2 | 84.7 |
| 0.5 | 59.0 | 6.7 | 86.0 | 83.7 | 52.8 | 53.7 | 45.6 | 84.7 |
| 0.6 | 59.1 | 4.4 | 86.1 | 83.5 | 52.8 | 54.7 | 47.2 | 84.7 |
| 0.7 | 59.6 | 5.6 | 85.1 | 84.0 | 53.6 | 55.8 | 47.6 | 85.3 |
| 0.8 | 59.4 | 5.6 | 84.9 | 84.5 | 53.1 | 53.6 | 48.8 | 85.3 |
| 0.9 | 59.8 | 3.3 | 85.0 | 84.7 | 53.0 | 56.0 | 49.4 | 87.0 |
| 1.0 | 59.5 | 3.3 | 84.7 | 85.1 | 53.0 | 54.9 | 49.4 | 85.8 |
| 1.2 | 59.1 | 0.0 | 84.8 | 86.0 | 52.6 | 56.6 | 49.1 | 84.3 |
| 1.5 | 59.9 | 3.3 | 84.6 | 84.9 | 52.4 | 55.9 | 50.5 | 87.3 |
| 2.0 | 57.9 | 0.0 | 83.2 | 84.2 | 50.2 | 52.0 | 48.1 | 87.7 |

**Table 9:** Detailed results on the math benchmarks for Table 1. Three times independent training and three times evaluation average are reported.

| Model | Method | AIME24 | GSM8K (8-shot) | GSM8K (0-shot) | MATH | MATH 500 | Minerva Math | SVAMP | Math-7 |
|---|---|---|---|---|---|---|---|---|---|
| Llama-3.2-1B | Vanilla | 1.1 | 46.9 | 1.8 | 15.8 | 15.1 | 20.3 | 65.3 | 23.8 |
| | FT (*full*) | 1.1 | 46.8 | 0.8 | 18.6 | 19.1 | 19.0 | 66.5 | 24.5 |
| | Shadow-FT (*full*) | 0.0 | 47.2 | 1.0 | 18.9 | 18.3 | 23.1 | 67.9 | 25.2 |
| | FT (*LoRA*) | 1.1 | 45.2 | 2.6 | 21.8 | 20.3 | 18.7 | 67.5 | 25.3 |
| | Shadow-FT (*LoRA*) | 0.0 | 47.8 | 4.6 | 22.1 | 24.5 | 25.1 | 66.4 | 27.2 |
| Llama-3.2-3B | Vanilla | 4.4 | 76.6 | 79.5 | 45.8 | 47.9 | 36.1 | 84.7 | 53.6 |
| | FT (*full*) | 8.9 | 77.0 | 77.0 | 45.2 | 44.2 | 32.4 | 84.1 | 52.7 |
| | Shadow-FT (*full*) | 8.9 | 77.8 | 80.4 | 47.2 | 47.9 | 37.6 | 84.8 | 54.9 |
| | FT (*LoRA*) | 4.4 | 76.5 | 73.5 | 46.4 | 47.7 | 31.7 | 83.1 | 51.9 |
| | Shadow-FT (*LoRA*) | 11.1 | 78.1 | 77.6 | 49.8 | 52.0 | 39.3 | 85.4 | 56.2 |
| Llama-3.1-8B | Vanilla | 6.7 | 83.6 | 85.0 | 49.1 | 49.2 | 40.8 | 83.2 | 56.8 |
| | FT (*full*) | 1.1 | 84.0 | 85.4 | 51.0 | 51.5 | 39.0 | 85.8 | 56.8 |
| | Shadow-FT (*full*) | 6.7 | 85.0 | 84.0 | 52.2 | 53.2 | 43.8 | 86.3 | 58.7 |
| | FT (*LoRA*) | 6.7 | 83.8 | 83.8 | 50.2 | 52.5 | 41.3 | 86.5 | 57.8 |
| | Shadow-FT (*LoRA*) | 6.7 | 85.0 | 84.5 | 52.0 | 53.0 | 48.3 | 86.0 | 59.4 |
| Qwen-3-4B | Vanilla | 18.9 | 87.8 | 92.2 | 70.3 | 82.3 | 73.4 | 91.5 | 73.8 |
| | FT (*full*) | 14.4 | 88.1 | 91.6 | 70.1 | 82.4 | 72.1 | 91.2 | 72.9 |
| | Shadow-FT (*full*) | 16.7 | 87.4 | 92.3 | 70.0 | 84.3 | 73.3 | 91.7 | 73.7 |
| | FT (*LoRA*) | 18.9 | 84.2 | 91.6 | 68.1 | 77.3 | 67.4 | 90.7 | 71.2 |
| | Shadow-FT (*LoRA*) | 28.9 | 88.3 | 92.5 | 70.4 | 84.5 | 73.8 | 92.8 | 75.9 |
| Qwen-3-8B | Vanilla | 22.2 | 87.3 | 93.4 | 70.8 | 83.1 | 73.2 | 91.6 | 74.5 |
| | FT (*full*) | 22.2 | 86.2 | 93.1 | 70.6 | 80.7 | 72.7 | 92.1 | 74.0 |
| | Shadow-FT (*full*) | 32.2 | 87.5 | 93.3 | 70.6 | 82.9 | 73.2 | 91.4 | 75.9 |
| | FT (*LoRA*) | 17.8 | 83.6 | 92.1 | 68.9 | 77.3 | 68.4 | 90.7 | 71.3 |
| | Shadow-FT (*LoRA*) | 22.2 | 88.5 | 92.9 | 70.5 | 84.1 | 73.6 | 91.8 | 74.8 |
| Qwen-3-14B | Vanilla | 20.0 | 90.0 | 95.3 | 72.1 | 85.2 | 75.7 | 92.6 | 75.8 |
| | FT (*full*) | 17.8 | 88.9 | 94.9 | 72.2 | 85.5 | 75.5 | 91.3 | 75.1 |
| | Shadow-FT (*full*) | 40.0 | 90.7 | 95.2 | 71.7 | 86.3 | 76.0 | 92.7 | 78.9 |
| | FT (*LoRA*) | 14.4 | 87.3 | 94.5 | 71.7 | 81.3 | 72.8 | 91.0 | 73.3 |
| | Shadow-FT (*LoRA*) | 36.7 | 90.7 | 95.9 | 71.3 | 86.7 | 76.1 | 93.2 | 78.7 |
| Qwen-2.5-32B | Vanilla | 16.7 | 84.3 | 95.5 | 78.0 | 83.1 | 71.7 | 89.3 | 74.1 |
| | FT (*full*) | 21.1 | 86.6 | 95.4 | 74.8 | 82.9 | 76.8 | 92.1 | 75.7 |
| | Shadow-FT (*full*) | 13.3 | 85.0 | 95.5 | 76.8 | 84.1 | 78.0 | 91.3 | 74.9 |
| | FT (*LoRA*) | 14.4 | 85.7 | 95.3 | 73.6 | 83.8 | 75.0 | 92.1 | 74.3 |
| | Shadow-FT (*LoRA*) | 18.9 | 86.3 | 95.6 | 76.0 | 84.3 | 77.3 | 91.3 | 75.7 |

**Table 10:** Detailed data on the code benchmarks for Table 1. Three times independent training and three times evaluation averages are reported.

| Model | Method | HumanEval | HumanEval$^+$ | LiveCodeBench | | | | Code-3 |
|---|---|---|---|---|---|---|---|---|
| | | | | Exec | Gen | Out | Avg | |
| Llama 3.2-1B | Vanilla | 40.9 | 35.0 | 4.0 | 7.0 | 0.2 | 3.7 | 26.5 |
| | FT (*full*) | 38.2 | 34.2 | 9.3 | 6.9 | 1.2 | 5.8 | 26.1 |
| | Shadow-FT (*full*) | 42.9 | 36.8 | 5.9 | 7.3 | 1.1 | 4.8 | 28.2 |
| | FT (*LoRA*) | 39.2 | 34.4 | 11.0 | 6.5 | 0.8 | 6.1 | 26.6 |
| | Shadow-FT (*LoRA*) | 41.9 | 35.4 | 10.5 | 6.9 | 2.2 | 6.5 | 27.9 |
| Llama 3.2-3B | Vanilla | 60.0 | 52.2 | 0.0 | 16.8 | 0.7 | 5.8 | 39.3 |
| | FT (*full*) | 57.9 | 53.7 | 4.5 | 16.3 | 6.2 | 9.0 | 40.2 |
| | Shadow-FT (*full*) | 60.6 | 54.1 | 0.0 | 16.8 | 2.1 | 6.3 | 40.3 |
| | FT (*LoRA*) | 59.1 | 50.8 | 9.7 | 17.1 | 15.5 | 14.1 | 41.4 |
| | Shadow-FT (*LoRA*) | 61.2 | 55.3 | 14.4 | 16.0 | 5.8 | 12.1 | 42.9 |
| Llama 3.1-8B | Vanilla | 69.7 | 62.8 | 17.3 | 19.8 | 23.2 | 20.1 | 50.9 |
| | FT (*full*) | 70.7 | 67.3 | 16.9 | 22.3 | 27.5 | 22.2 | 53.4 |
| | Shadow-FT (*full*) | 70.1 | 63.6 | 16.6 | 20.8 | 27.8 | 21.7 | 51.8 |
| | FT (*LoRA*) | 70.7 | 63.4 | 16.3 | 21.0 | 26.8 | 21.4 | 51.8 |
| | Shadow-FT (*LoRA*) | 71.1 | 50.4 | 14.9 | 21.3 | 27.3 | 21.2 | 50.9 |
| Qwen-3-4B | Vanilla | 77.9 | 71.3 | 41.8 | 48.8 | 59.7 | 50.1 | 66.4 |
| | FT (*full*) | 80.9 | 70.9 | 43.1 | 46.1 | 53.0 | 47.4 | 66.4 |
| | Shadow-FT (*full*) | 80.3 | 71.1 | 42.5 | 49.7 | 60.1 | 50.8 | 67.4 |
| | FT (*LoRA*) | 76.4 | 69.1 | 13.1 | 41.1 | 45.6 | 33.3 | 59.6 |
| | Shadow-FT (*LoRA*) | 81.3 | 76.8 | 43.2 | 49.1 | 60.6 | 51.0 | 69.7 |
| Qwen-3-8B | Vanilla | 85.8 | 79.9 | 42.3 | 51.3 | 63.4 | 52.3 | 72.7 |
| | FT (*full*) | 82.7 | 79.3 | 42.9 | 51.8 | 59.7 | 51.5 | 71.2 |
| | Shadow-FT (*full*) | 86.8 | 79.3 | 41.9 | 52.3 | 65.2 | 53.1 | 73.1 |
| | FT (*LoRA*) | 84.2 | 78.5 | 42.0 | 45.7 | 50.9 | 46.2 | 69.6 |
| | Shadow-FT (*LoRA*) | 84.6 | 77.6 | 41.9 | 52.4 | 66.1 | 53.5 | 71.9 |
| Qwen-3-14B | Vanilla | 86.8 | 83.1 | 51.9 | 55.8 | 74.2 | 60.6 | 76.8 |
| | FT (*full*) | 87.6 | 83.5 | 50.9 | 54.3 | 67.3 | 57.5 | 76.2 |
| | Shadow-FT (*full*) | 87.4 | 82.9 | 52.1 | 55.6 | 74.4 | 60.7 | 77.0 |
| | FT (*LoRA*) | 85.6 | 82.3 | 51.2 | 51.3 | 62.5 | 55.0 | 74.4 |
| | Shadow-FT (*LoRA*) | 87.8 | 84.4 | 50.7 | 56.8 | 76.4 | 61.3 | 77.8 |
| Qwen-2.5-32B | Vanilla | 86.4 | 82.1 | 58.3 | 54.6 | 64.6 | 59.1 | 75.9 |
| | FT (*full*) | 85.6 | 81.1 | 60.3 | 55.8 | 66.4 | 60.8 | 75.8 |
| | Shadow-FT (*full*) | 86.6 | 81.5 | 60.5 | 55.7 | 64.0 | 60.1 | 76.1 |
| | FT (*LoRA*) | 85.4 | 81.7 | 60.9 | 55.0 | 64.8 | 60.7 | 75.9 |
| | Shadow-FT (*LoRA*) | 87.4 | 80.5 | 61.8 | 55.0 | 64.9 | 60.6 | 76.2 |

**Table 11:** Detailed results on the general Reasoning benchmarks for Table 1.

| Method | MMLU | MMLU Pro | WinoG | DROP | ARC Challenge | BBH (0-shot) | BBH (3-shot) | GPQA Diamond | TheoremQA | Knowledge-9 |
|---|---|---|---|---|---|---|---|---|---|---|
| | | | | | *Llama-3.2-1B* | | | | | |
| Vanilla | 46.8 | 21.4 | 51.9 | 42.7 | 56.6 | 24.4 | 26.1 | 27.8 | 9.9 | 34.2 |
| FT (*full*) | 46.9 | 21.8 | 50.4 | 39.0 | 56.6 | 20.4 | 24.8 | 24.8 | 10.8 | 32.8 |
| Shadow-FT (*full*) | 47.1 | 22.7 | 51.1 | 41.6 | 52.9 | 22.2 | 20.8 | 26.3 | 9.6 | 32.7 |
| FT (*LoRA*) | 46.7 | 22.1 | 51.2 | 40.8 | 56.6 | 20.9 | 26.3 | 23.2 | 11.8 | 33.3 |
| Shadow-FT (*LoRA*) | 46.6 | 23.2 | 51.4 | 43.9 | 52.2 | 17.2 | 20.4 | 25.3 | 10.6 | 32.3 |
| | | | | | *Llama-3.2-3B* | | | | | |
| Vanilla | 62.4 | 39.7 | 53.9 | 71.8 | 78.6 | 41.8 | 49.2 | 29.3 | 17.4 | 49.3 |
| FT (*full*) | 62.0 | 39.2 | 54.5 | 71.7 | 79.0 | 41.8 | 51.8 | 25.8 | 18.5 | 49.4 |
| Shadow-FT (*full*) | 62.4 | 40.4 | 54.3 | 72.1 | 79.0 | 41.7 | 50.0 | 28.3 | 17.6 | 49.5 |
| FT (*LoRA*) | 61.9 | 39.9 | 51.1 | 71.7 | 82.7 | 41.4 | 49.4 | 25.3 | 18.4 | 49.1 |
| Shadow-FT (*LoRA*) | 62.1 | 41.6 | 54.6 | 72.0 | 79.0 | 38.6 | 49.6 | 26.8 | 16.1 | 48.9 |
| | | | | | *Llama-3.1-8B* | | | | | |
| Vanilla | 69.5 | 48.5 | 59.4 | 81.4 | 85.4 | 44.6 | 67.6 | 25.8 | 27.3 | 56.6 |
| FT (*full*) | 69.7 | 49.2 | 60.9 | 80.0 | 87.1 | 46.8 | 71.1 | 30.8 | 30.6 | 58.5 |
| Shadow-FT (*full*) | 69.6 | 49.3 | 60.2 | 81.7 | 85.8 | 46.8 | 67.0 | 28.3 | 29.5 | 57.6 |
| FT (*LoRA*) | 69.3 | 48.9 | 60.0 | 79.5 | 86.4 | 48.8 | 68.0 | 30.3 | 26.8 | 57.5 |
| Shadow-FT (*LoRA*) | 69.4 | 50.8 | 60.2 | 80.1 | 85.4 | 51.6 | 68.8 | 32.8 | 29.1 | 58.7 |
| | | | | | *Qwen-3-4B* | | | | | |
| Vanilla | 70.7 | 57.1 | 57.7 | 77.3 | 91.5 | 57.7 | 78.7 | 37.4 | 44.6 | 63.6 |
| FT (*full*) | 70.7 | 54.2 | 56.8 | 75.9 | 91.2 | 57.3 | 77.2 | 38.9 | 44.4 | 63.0 |
| Shadow-FT (*full*) | 71.4 | 57.0 | 57.4 | 77.7 | 92.2 | 58.4 | 78.4 | 45.0 | 46.5 | 64.9 |
| FT (*LoRA*) | 71.9 | 51.2 | 59.0 | 69.1 | 91.5 | 54.7 | 73.5 | 39.4 | 39.9 | 61.1 |
| Shadow-FT (*LoRA*) | 71.8 | 58.2 | 58.8 | 79.1 | 91.9 | 59.6 | 77.0 | 46.0 | 42.4 | 65.0 |
| | | | | | *Qwen-3-8B* | | | | | |
| Vanilla | 76.5 | 55.8 | 55.7 | 85.2 | 91.9 | 59.8 | 80.0 | 46.5 | 31.0 | 64.7 |
| FT (*full*) | 76.3 | 53.0 | 54.8 | 84.8 | 91.2 | 60.1 | 80.1 | 44.4 | 36.5 | 64.6 |
| Shadow-FT (*full*) | 76.6 | 56.0 | 54.8 | 85.8 | 92.2 | 59.2 | 79.8 | 53.5 | 32.1 | 65.6 |
| FT (*LoRA*) | 76.1 | 57.2 | 55.9 | 80.6 | 92.5 | 59.0 | 75.5 | 41.4 | 40.9 | 64.3 |
| Shadow-FT (*LoRA*) | 78.6 | 61.5 | 55.0 | 85.8 | 92.5 | 59.3 | 79.6 | 56.6 | 41.1 | 67.8 |
| | | | | | *Qwen-3-14B* | | | | | |
| Vanilla | 79.4 | 64.2 | 68.5 | 86.3 | 94.6 | 61.4 | 84.2 | 47.0 | 54.6 | 71.1 |
| FT (*full*) | 79.7 | 61.3 | 67.8 | 85.5 | 94.9 | 61.2 | 84.1 | 47.5 | 53.0 | 70.6 |
| Shadow-FT (*full*) | 79.6 | 64.9 | 68.7 | 86.9 | 94.6 | 60.3 | 83.9 | 46.5 | 57.6 | 71.4 |
| FT (*LoRA*) | 79.6 | 60.7 | 68.5 | 84.0 | 95.3 | 63.3 | 83.0 | 47.0 | 51.9 | 70.4 |
| Shadow-FT (*LoRA*) | 79.8 | 66.1 | 69.1 | 88.1 | 93.6 | 58.2 | 83.6 | 48.0 | 56.8 | 71.5 |
| | | | | | *Qwen-2.5-32B* | | | | | |
| Vanilla | 83.4 | 68.8 | 82.2 | 88.1 | 95.3 | 63.6 | 84.6 | 39.9 | 54.3 | 73.4 |
| FT (*full*) | 83.4 | 68.3 | 81.9 | 88.7 | 94.6 | 63.0 | 83.8 | 42.4 | 56.5 | 73.6 |
| Shadow-FT (*full*) | 83.2 | 69.1 | 82.6 | 88.4 | 95.6 | 64.3 | 82.9 | 39.4 | 55.8 | 73.5 |
| FT (*LoRA*) | 83.6 | 68.9 | 82.2 | 88.8 | 94.9 | 62.8 | 83.7 | 44.4 | 54.8 | 73.8 |
| Shadow-FT (*LoRA*) | 83.0 | 68.8 | 82.7 | 88.6 | 94.6 | 64.3 | 83.4 | 43.9 | 54.8 | 73.8 |

**Table 12:** Case study example from MATH-500. Shadow-FT can avoid *errors* with the knowledge learned from the base model, while the vanilla tune method fails. We extract the key step for better visualization.

| | |
|---|---|
| **Input** | ```Question:```
```How many positive two-digit integers are```
```factors of both 100 and 150?``` |
| **Ground Truth** | ```$\boxed {3}$ (The common two-digit factors```
```are 10, 25, and 50).``` |
| **Vanilla FT** | ```...```
```1.  **Find the GCD of 100 and 150:**```
```...```
```2.  **Identify the two-digit factors of```
```50:**```
```...  stop counting at 25.```
```Thus, the positive two-digit integers that```
```are factors of both 100 and 150 are 10 and```
```25.```
```$\boxed {2}$``` |
| **Shadow-FT (Ours)** | ```...```
```###### Step 1:  Find the Greatest Common```
```Divisor (GCD) of 100 and 150```
```...```
```###### Step 2:  Find All Factors of the GCD```
```(50)```
```...```
```###### Step 3:  Identify the Two-Digit```
```Factors```
```...```
```(Note:  50 is a two-digit number too, but```
```we'll check it as well.)```
```So the two-digit factors of 50 are:```
```$$10, 25, 50$$```
```###### Step 4:  Count the Two-Digit Factors```
```...```
```###### Final Answer:```
```$$\boxed {3}$$``` |

## E   CASE STUDY ON LLAMA-3.1-8B

Table 12 presents a representative case from Math-500 benchmark generated by Llama-3.1-8B-Instruct tuned via vanilla FT and Shadow-FT, respectively. While the vanilla fine-tuned model partially solves the problem, it stops prematurely and misses one of the valid two-digit factors, resulting in an incorrect prediction of $\boxed {2}$. In contrast, Shadow-FT correctly finds the GCD, enumerates all factors, and recognizes that 50 is also a two-digit factor, producing the correct answer $\boxed {3}$. This example highlights the capability of Shadow-FT to leverage knowledge from the BASE model and reason more comprehensively.

## F   LIMITATIONS

In Shadow-FT, we first tune the BASE model and then graft the weight updates to the INSTRUCT model. However, there are some LLMs for which the paired BASE models are not available, such as Qwen3-32B and Qwen3-Next. For these LLMs, we can not apply Shadow-FT. Therefore, finding a proper 'shadow' for these models is an interesting topic for future work.

