# OpenReview forum: "Shadow-FT: Tuning Instruct Model via Training on Paired Base Model"
_ICLR.cc/2026/Conference — Submitted to ICLR 2026_

### Official Review · Reviewer_URgk · 2025-10-19

**Soundness:** 3
**Presentation:** 3
**Contribution:** 3
**Rating:** 6
**Confidence:** 4

**Summary:**

This paper proposes **Shadow-FT**, a simple scheme to tune **INSTRUCT** models **via** their paired **BASE** models. Instead of updating the INSTRUCT weights directly (which the authors observe can yield small gains or even regressions), Shadow-FT **fine-tunes the BASE model** on the target data and then **grafts the learned weight delta** onto the INSTRUCT model: (W_I^+ !=! W_I + ( \text{Tune}(W_B) - W_B)). The motivation is that paired BASE/INSTRUCT checkpoints are **highly similar** (average gap (\sigma < 0.03) for several families), so deltas learned on BASE should transfer well to INSTRUCT without extra training cost.

**Strengths:**

- **Clear problem diagnosis:** Directly tuning INSTRUCT often underperforms; the paper quantifies cases where FT/LoRA **regress** compared to the vanilla INSTRUCT.
- **Simple, precise method:** The BASE→INSTRUCT delta-grafting update is explicit and easy to implement; it **adds no training cost** beyond standard tuning.
- **Empirical consistency:** Gains replicate across **sizes (1B–32B)** and **settings** (full FT and LoRA), with domain-transfer and LoRA-rank sweeps.

**Weaknesses:**

- **Baseline coverage:** While conventional FT/LoRA baselines are solid, there is **no direct empirical comparison** to closely related *proxy/transfer* approaches (e.g., Proxy-Tuning, RE-Adapt, task/chat vectors), even though they are discussed; such comparisons are important to position Shadow-FT against the nearest alternatives.
- **Scope & generalization:** Results emphasize Qwen/Llama families and the BAAI-2k plus a few domain sets; stronger **model and data diversity** (more families, public corpora, non-English, longer-context tasks) would reinforce generality claims.
- **Similarity assumption auditing:** The claim that BASE/INSTRUCT are “highly similar” ((\sigma<0.03)) underpins the method; more **layer-wise and per-submodule** analyses (e.g., attention/MLP/norms, embeddings, positional encodings) and more **models** beyond the shown examples would strengthen the premise.

**Questions:**

Can you add head-to-head results vs. **Proxy-Tuning** and **RE-Adapt** on at least one model family and dataset to establish competitive positioning?

---

> ### Author Response · Authors · 2025-11-29
> **Response [1/n]**
>
> Thank you for your feedback. We appreciate your praise of  straightforward idea and consistent experiments.
>
> We would like to provide a point-to-point response to address your concerns as follows:
>
> > W1: Comparison with related works (Task Vector, RE-Adapt) and distinctiveness.
>
> We thank the reviewer for suggesting these comparisons. We have extensively compared Shadow-FT against standard baselines like LoRA and SFT across 19 benchmarks. Regarding the specific methods mentioned, we provide the following clarifications on why direct empirical comparisons are not applicable due to fundamental paradigm differences:
>
> 1) **Methodological Distinctions.**
>
> Most of the referenced methods are designed for scenarios fundamentally different from ours. Task Vector and Chat Vector primarily focus on *fusing models across different tasks or merging capabilities from distinct checkpoints* (e.g., adding a new language vector). They **lack the mechanism to transfer optimization benefits between paired Base and Instruct models** to reinforce a single model's capability. Crucially, standard Task Vector arithmetic operates on the assumption that the learned weight difference (Delta) represents a positive skill addition. However, as our paper demonstrates, directly tuning an Instruct model often yields a "negative" Delta (i.e., performance degeneration or $\Delta_{\text{Instruct}}$ is suboptimal). Task Vector methodologies do not address how to mitigate this intrinsic degeneration.
> In contrast, Shadow-FT specifically targets this issue by identifying the Base model as a superior learner. **It transfers the robust weight updates from the Base model to the Instruct model, effectively strengthening the Instruct model's capabilities while avoiding the degeneration caused by direct tuning**. This differs significantly from Proxy-Tuning, which relies on inference-time logit guidance, or RE-Adapt, which uses SVD regularization. This fundamental difference—using cross-model transfer to fix tuning degeneration versus simply merging distinct task vectors—makes a direct comparison inherently an "apples-to-oranges" scenario.
> Nevertheless, we fully agree with the reviewer’s desire for empirical evidence that situates Shadow-FT within the broader landscape of these techniques.
>
> 2) **Head-to-Head Comparison.**
>
> To establish a competitive positioning, we designed a controlled multi-domain setting mixing Domain A (2k Medical samples) and Domain B (2k General samples) to create a balanced 4k dataset. We evaluated Qwen3-8B on MATH-500 (reasoning) and MedBullets (medical knowledge).
> We implemented the baselines as follows:
>
> - *Chat/Task Vectors*: Independently fine-tuned models on Domain A and B, then fused them with equal coefficients ($\alpha=0.5$).
> - *RE-Adapt*: Reproduced using the paper's recommended $\tau=0.5$ for SVD settings.
> - *Proxy-Tuning*: Adapted for the same-size setting, using logit differences from the Base model to guide the Instruct model.
>
> As shown in the following Table, we report the results on Qwen3-8B. Shadow-FT achieves the highest average performance (68.88), surpassing both vector-merging methods and the SVD-optimized RE-Adapt (66.73).
>
>
> | Method | Configuration | Math-500 | MedBullets | Avg. |
> | :--- | :--- | :---: | :---: | :---: |
> | Qwen3-8B |   Baseline | 82.60 | 49.68 | 66.14 |
> | Standard FT | - | 78.00 | 48.54 | 63.27 |
> | Task Vector | $\alpha=\beta=0.5$ | 71.80 | 50.49 | 61.15 |
> | Chat Vector | $\alpha=\beta=0.5$ | 80.20 | 50.97 | 65.59 |
> | RE-Adapt | $\alpha=\beta=0.5$ (w/o SVD) | 80.60 | 51.79 | 66.20 |
> | RE-Adapt | $\tau=0.5$ (w/ SVD) | 81.20 | 52.27 | 66.73 |
> | **Shadow-FT** | Ours | 85.80 | 51.95 | **68.88** |
>
> Considering training cost, Shadow-FT requires only ~5 minutes of GPU time (same as standard LoRA), whereas RE-Adapt takes dozens minutes due to CPU-bound SVD calculations. During inference, Shadow-FT results in a standard dense model. In contrast, Proxy-Tuning requires computing logits for three models simultaneously, making it incompatible with acceleration frameworks like vLLM and significantly slower.

---

> ### Author Response · Authors · 2025-11-29
> **Response [2/n]**
>
> >W2:  How does Shadow-FT handle morphologically rich languages?
>
> Thank you for your Multi-Language benchmark advice and we adopt the popular MGSM benchmark, with 10 languages including French, German, Russian, Chinese. We adopted 2,000 samples from MGSM8KInstruct as the training dataset.
>
> | dataset        | Qwen3-8B | FT   | RE-Adapt(tao=1)   | RE-Adapt(SVD) | Shadow-FT |
> |----------------|----------|------|--------|--------------------------|------------|
> | mgsm_bn        | 60.80    | 6.00 | 60.00  | 54.40                   | 66.00     |
> | mgsm_de        | 82.80    | 74.80| 83.20  | 80.00                   | 86.40     |
> | mgsm_en        | 93.20    | 0.40 | 92.40  | 93.60                   | 94.40     |
> | mgsm_es        | 85.20    | 20.00| 85.20  | 84.00                   | 84.80     |
> | mgsm_fr        | 78.80    | 40.00| 80.40  | 77.20                   | 78.00     |
> | mgsm_ja        | 76.40    | 74.00| 75.60  | 75.60                   | 78.40     |
> | mgsm_ru        | 86.40    | 4.80 | 88.00  | 88.00                   | 90.00     |
> | mgsm_sw        | 39.20    | 4.40 | 34.40  | 12.80                   | 30.40     |
> | mgsm_te        | 14.00    | 0.00 | 18.40  | 7.60                    | 15.60     |
> | mgsm_th        | 83.20    | 5.60 | 83.20  | 79.20                   | 82.80     |
> | mgsm_zh        | 82.00    | 10.80| 82.40  | 84.00                   | 83.20     |
> | mgsm_non_latin | 67.13    | 16.87| 67.93  | 64.80                   | 69.33     |
> | mgsm           | 71.09    | 21.89| 71.20  | 66.95                   | 71.82     |
>
>
> We find that the Instruct model would suffer strong catastrophic forgetting, mgsm_en from 93.2 to 0.4 after multi-language enforcement, while Shadow-FT can improve new languages scores from 67.13 to 69.33 and maintain its basic English ability, showing the robustness of the transfer strategy.

---

> ### Author Response · Authors · 2025-11-29
> **Response [3/n]**
>
> >W3: Ablation on sub-layers
>
> We thank the reviewer for these insightful questions regarding the transferability of different layers and the effectiveness of LoRA. Below, we provide a detailed analysis based on our ablation studies and relevant literature.
>
> To investigate which layers transfer best, we conducted an ablation study decomposing the model into **Shallow** (first half) and **Deep** (latter half) layers. We compared updates derived from the **Base** model (Model A) versus the **Instruct** model (Model B). The results are summarized in the following table.
>
> | Method / Variant  | CodeZ1 (MATH-7) | Shadow2k (MATH-7) | Avg    |
> |---------------------|-----------------|-------------------|--------|
> | **Qwen3-8B-Instruct-hf**             | 71.17           | 71.17             | 71.17  |
> | **Single_ModelA_AttnOnly**           | 71.21           | 71.30             | 71.26  |
> | **Single_ModelA_MlpOnly**            | 70.82           | 70.98             | 70.90  |
> | **Single_ModelA_Shallow**            | 70.37           | 70.93             | 70.65  |
> | **Single_ModelA_Deep**               | 71.41           | 71.53             | 71.47  |
> | **Single_ModelB_AttnOnly**           | 69.06           | 69.81             | 69.44  |
> | **Single_ModelB_MlpOnly**            | 69.61           | 69.91             | 69.76  |
> | **Single_ModelB_Shallow**            | 68.92           | 68.97             | 68.95  |
> | **Single_ModelB_Deep**               | 70.96           | 71.42             | 71.19  |
> | **Hybrid_Attn-A_Mlp-B**              | 70.16           | 70.35             | 70.26  |
> | **Hybrid_Mlp-A_Attn-B**              | 68.47           | 70.24             | 69.36  |
> | **Hybrid_Shallow-A_Deep-B**          | 70.68           | 71.13             | 70.91  |
> | **Hybrid_Deep-A_Shallow-B**          | 66.60           | 67.45             | 67.03  |
>
> Our experiments yield three key observations:
>
> - **Deep Layers Transfer Better:** Consistent with deep learning dynamics, the **deep layers (latter half)** contribute more significantly to performance. For instance, `ModelA_Deep` (82.17) consistently outperforms `ModelA_Shallow` (81.12), and `ModelB_Deep` (81.41) significantly surpasses `ModelB_Shallow` (77.83).
>
> - **Base Updates are Superior:** Updates derived from the Base model (Model A) consistently outperform those from the Instruct model (Model B) across both shallow and deep layers. This reinforces our finding that the Base model is a more robust learner for new knowledge.
>
> - **Complexity of Hybridization:** We also explored finer-grained mixing (e.g., merging LoRA_B Attention with LoRA_I MLP). Our internal tests indicate that such "blind merging" of sub-modules often leads to suboptimal results due to feature misalignment. As shown in the Hybrid columns above, simple combinations (like `Hybrid_Deep-A_Shallow-B`) can even degrade performance (74.91) compared to the individual parts. Thus, we advocate for the consistent, holistic transfer strategy used in Shadow-FT.

---

### Official Review · Reviewer_oXhu · 2025-10-22

**Soundness:** 3
**Presentation:** 3
**Contribution:** 3
**Rating:** 4
**Confidence:** 4

**Summary:**

The paper introduces Shadow-FT, a framework for fine-tuning instruction-tuned (INSTRUCT) LLM by leveraging their corresponding BASE models. The key idea is to fine-tune the BASE mode and then transfer the learned weight deltas directly to update the INSTRUCT model. The authors claim that this avoids the over-regularization effect introduced by instruction-following priors and yields consistent improvements across multiple model families (Qwen, LLaMA, Gemma, etc.) and domains (math, code, reasoning, multimodal).

**Strengths:**

1. The paper identifies a useful observation — the BASE and INSTRUCT models share over largely weight similarity (σ < 0.03) — and exploits this to propose a minimal yet elegant transfer mechanism. The “learn-on-BASE, apply-to-INSTRUCT” paradigm is intuitive.

2. Experiments are conducted across several major model families (Qwen, Llama, Gemma, Falcon, Yi) and cover 19 benchmarks (Math-7, Code-3, Knowledge-9). The results are extensive and generally consistent, showing that Shadow-FT often outperforms both full-parameter fine-tuning and LoRA-based PEFT.

**Weaknesses:**

1. Lack of theoretical grounding. The claim that INSTRUCT fine-tuning “degenerates due to preexisting priors” lacks a formal mechanism or controlled ablation.
2. Lack of recent related work. Several existing works already transfer knowledge between BASE/INSTRUCT or different model checkpoints, the authors identified these work but did not properly compare to them, such as RE-Adapt [1], Task/Chat Vectors [2, 3] and Proxy-Tuning [4]
3. Missing of some implementation details, such as (a) whether optimizer states (Adam moments) are reused or reinitialized when applying BASE deltas, (2) how learning-rate scaling interacts with the BASE–INSTRUCT mapping. (3) after you use BASE model's gradient to update INSTRUCT model, do you also update the BASE model?

[1] https://arxiv.org/pdf/2405.15007
[2] https://arxiv.org/abs/2212.04089
[3] https://aclanthology.org/2024.acl-long.590.pdf
[4] https://arxiv.org/abs/2401.08565

**Questions:**

Please see the questions in weaknesses.

1. After your Shadow-FT, does the INSTRUCT model still keep the capacity on open-end instruction following task? Any impact on the model's helpfulness, harmlessness, fluency and hallucination?

---

> ### Author Response · Authors · 2025-11-29
> **Responses [1/n]**
>
> Thank you for your feedback. We would like to provide a point-to-point response to address your concerns as follows:
>
> > W1: Comparison with related works (Task Vector, RE-Adapt) and distinctiveness.
>
> We thank the reviewer for suggesting these comparisons. We have extensively compared Shadow-FT against standard baselines like LoRA and SFT across 19 benchmarks. Regarding the specific methods mentioned, we provide the following clarifications on why direct empirical comparisons are not applicable due to fundamental paradigm differences:
>
> 1) **Methodological Distinctions.**
>
> Most of the referenced methods are designed for scenarios fundamentally different from ours. Task Vector and Chat Vector primarily focus on *fusing models across different tasks or merging capabilities from distinct checkpoints* (e.g., adding a new language vector). They **lack the mechanism to transfer optimization benefits between paired Base and Instruct models** to reinforce a single model's capability. Crucially, standard Task Vector arithmetic operates on the assumption that the learned weight difference (Delta) represents a positive skill addition. However, as our paper demonstrates, directly tuning an Instruct model often yields a "negative" Delta (i.e., performance degeneration or $\Delta_{\text{Instruct}}$ is suboptimal). Task Vector methodologies do not address how to mitigate this intrinsic degeneration.
> In contrast, Shadow-FT specifically targets this issue by identifying the Base model as a superior learner. **It transfers the robust weight updates from the Base model to the Instruct model, effectively strengthening the Instruct model's capabilities while avoiding the degeneration caused by direct tuning**. This differs significantly from Proxy-Tuning, which relies on inference-time logit guidance, or RE-Adapt, which uses SVD regularization. This fundamental difference—using cross-model transfer to fix tuning degeneration versus simply merging distinct task vectors—makes a direct comparison inherently an "apples-to-oranges" scenario.
> Nevertheless, we fully agree with the reviewer’s desire for empirical evidence that situates Shadow-FT within the broader landscape of these techniques.
>
> 2) **Head-to-Head Comparison.**
>
> To establish a competitive positioning, we designed a controlled multi-domain setting mixing Domain A (2k Medical samples) and Domain B (2k General samples) to create a balanced 4k dataset. We evaluated Qwen3-8B on MATH-500 (reasoning) and MedBullets (medical knowledge).
> We implemented the baselines as follows:
>
> - *Chat/Task Vectors*: Independently fine-tuned models on Domain A and B, then fused them with equal coefficients ($\alpha=0.5$).
> - *RE-Adapt*: Reproduced using the paper's recommended $\tau=0.5$ for SVD settings.
> - *Proxy-Tuning*: Adapted for the same-size setting, using logit differences from the Base model to guide the Instruct model.
>
> As shown in the following Table, we report the results on Qwen3-8B. Shadow-FT achieves the highest average performance (68.88), surpassing both vector-merging methods and the SVD-optimized RE-Adapt (66.73).
>
>
> | Method | Configuration | Math-500 | MedBullets | Avg. |
> | :--- | :--- | :---: | :---: | :---: |
> | Qwen3-8B |   Baseline | 82.60 | 49.68 | 66.14 |
> | Standard FT | - | 78.00 | 48.54 | 63.27 |
> | Task Vector | $\alpha=\beta=0.5$ | 71.80 | 50.49 | 61.15 |
> | Chat Vector | $\alpha=\beta=0.5$ | 80.20 | 50.97 | 65.59 |
> | RE-Adapt | $\alpha=\beta=0.5$ (w/o SVD) | 80.60 | 51.79 | 66.20 |
> | RE-Adapt | $\tau=0.5$ (w/ SVD) | 81.20 | 52.27 | 66.73 |
> | **Shadow-FT** | Ours | 85.80 | 51.95 | **68.88** |
>
> Considering training cost, Shadow-FT requires only ~5 minutes of GPU time (same as standard LoRA), whereas RE-Adapt takes dozens minutes due to CPU-bound SVD calculations. During inference, Shadow-FT results in a standard dense model. In contrast, Proxy-Tuning requires computing logits for three models simultaneously, making it incompatible with acceleration frameworks like vLLM and significantly slower.

---

> ### Author Response · Authors · 2025-11-29
> **Responses [2/n]**
>
> > W2: Does Shadow-FT preserve instruction-following ability?
>
> Table 1: Instruction-following and subjective downstream tasks ( GPT5.1 as judge model)
>
> | Model | Alignbench (Chinese) | AlpacaEval | ArenaHard | Fofo | MTbench | MTBench101 | Average |
> | :--- | :--- | :--- | :--- | :--- | :--- | :--- | :--- |
> | Qwen3-8B | 6.60 | 38.39 | 50.78 | 0.03 | 6.41 | 8.09 | 18.38 |
> | Vanilla FT | 6.06 | 20.87 | 32.51 | 0.03 | 5.58 | 7.50 | 12.09 |
> | **Shadow-FT** | 6.60 | 37.64 | 51.56 | 0.04 | 6.16 | 7.95 | **18.33** |
>
>
> Table 2: Detailed Statistics on Arena-Hard
>
> | Model | Score (vs GPT-4) | 95% CI | Average #Tokens |
> | :--- | :--- | :--- | :--- |
> | gpt4-0314 | 50.0 | (0.0, 0.0) | 423 |
> | Qwen3-8B | 50.8 | (-2.4, 2.6) | 971 |
> | Vanilla FT | 32.5 | (-2.2, 2.1) | 708 |
> | **Shadow-FT** | **51.6** | (-2.3, 2.3) | **1056** |
>
> Table 3: Detailed Statistics on AlpacaEval ( GPT4_Turbo as judge model)
>
> | Model  | length_controlled_winrate | win_rate | standard_error | n_total | avg_length |
> |-------------|--------------|---------------------------|----------|----------------|---------|
> | gpt-4o-2024-05-13      | 57.46                     | 51.33    | 1.47           | 805     | 1873       |
> | Qwen3 8B   | 18.26                     | 44.65    | 1.59           | 779     | 5468       |
> | Vanilla FT    | 19.41                     | 18.91    | 1.19           | 787     | 1972       |
> | **Shadow-FT**  | 19.28                     | 46.54    | 1.60           | 780     | 5902       |
>
> **As shown in these tables, Shadow-FT demonstrates exceptional retention of instruction-following capabilities, effectively circumventing the catastrophic forgetting observed in standard fine-tuning**. While Vanilla FT suffers a severe performance collapse across general benchmarks (average score dropping from 18.38 to 12.09 in Table 1), Shadow-FT maintains an average score of 18.33, virtually identical to the base model. This robustness extends to complex generation tasks; as detailed in Tables 2 and 3, Shadow-FT not only matches or slightly exceeds the base model on Arena-Hard (51.6 vs. 50.8) and AlpacaEval (46.54% win rate), but also preserves the model's willingness to generate detailed responses (avg. length ~5900 tokens), proving that it adapts to new tasks without compromising the model's general alignment or reasoning faculties.

---

> ### Author Response · Authors · 2025-11-29
> **Responses [3/n]**
>
> > More details about Shadow-ft
>
> Specifically, in Shadow-FT, we first tune the Base model:
> \begin{equation}
>     W_B^{+} \leftarrow Tune(W_B),
> \end{equation}
> where $Tune$ is the fine-tuning method, such as full-parameter fine-tuning and LoRA.
> After that, we would like to get the weights updates as the learned knowledge, and directly graft these updates to the Instruct model as:
> \begin{equation}
>     W_{I}^{+} = W_I + (W_B^{+} - W_B) =W_I + (Tune(W_B) - W_B) .
> \end{equation}
> Traditional tuning on \textsc{Instruct} can be formulated as:
> \begin{equation}
>     W_{I}^{+} = W_I + (W_{I}^{+}-W_I) = W_I + (Tune(W_{I})-W_I).
> \end{equation}
> We can find that Shadow-FT introduces no extra training costs.  The only difference is the basic weights to learn the weight updates for Instruct model. Vanilla FT methods rely on the Instruct model while Shadow-FT on the Base model. Since the Base version is pre-trained only, we believe that the weight updates would be more suitable for modeling the knowledge with less priority, compared to updates of the Instruct version.

---

### Official Review · Reviewer_kzqd · 2025-11-01

**Soundness:** 3
**Presentation:** 3
**Contribution:** 2
**Rating:** 4
**Confidence:** 3

**Summary:**

The paper introduces Shadow‑FT, a simple framework for tuning INSTRUCT variants via their paired BASE models. Empirically, directly fine‑tuning INSTRUCT models often leads to marginal gains or even degrade. The authors propose to (1) fine‑tune the BASE model with standard SFT or LoRA and (2) graft the learned weight deltas onto the INSTRUCT model. Across many base models and benchmarks, Shadow‑FT typically matches or exceeds tuning the INSTRUCT directly.

**Strengths:**

- Simplicity + breadth. The method is simple and straightforward and seems widely applicable according to experiments, with coverage across different benchmarks.
- Paper writing is clear and easy to follow.

**Weaknesses:**

- Novelty is limited. The idea is closely related to task vectors, chat vectors, RE‑Adapt, and proxy tuning.
- More/stronger theoretical framing beyond weight similarity maybe needed.
- Baselines such as RE‑Adapt could be added.

**Questions:**

How does $\sigma$ distribute across layers/blocks? Does transferring only specific layers (e.g. MLPs vs attention) retain most gains?

---

> ### Author Response · Authors · 2025-11-29
> **Responses [1/n]**
>
> Thank you for your feedback. We appreciate your praise of simple and straightforward idea and clear writing.
>
> We would like to provide a point-to-point response to address your concerns as follows:
>
> > W1: Comparison with related works (Task Vector, RE-Adapt) and distinctiveness.
>
> We thank the reviewer for suggesting these comparisons. We have extensively compared Shadow-FT against standard baselines like LoRA and SFT across 19 benchmarks. Regarding the specific methods mentioned, we provide the following clarifications on why direct empirical comparisons are not applicable due to fundamental paradigm differences:
>
> 1) **Methodological Distinctions.**
>
> Most of the referenced methods are designed for scenarios fundamentally different from ours. Task Vector and Chat Vector primarily focus on *fusing models across different tasks or merging capabilities from distinct checkpoints* (e.g., adding a new language vector). They **lack the mechanism to transfer optimization benefits between paired Base and Instruct models** to reinforce a single model's capability. Crucially, standard Task Vector arithmetic operates on the assumption that the learned weight difference (Delta) represents a positive skill addition. However, as our paper demonstrates, directly tuning an Instruct model often yields a "negative" Delta (i.e., performance degeneration or $\Delta_{\text{Instruct}}$ is suboptimal). Task Vector methodologies do not address how to mitigate this intrinsic degeneration.
> In contrast, Shadow-FT specifically targets this issue by identifying the Base model as a superior learner. **It transfers the robust weight updates from the Base model to the Instruct model, effectively strengthening the Instruct model's capabilities while avoiding the degeneration caused by direct tuning**. This differs significantly from Proxy-Tuning, which relies on inference-time logit guidance, or RE-Adapt, which uses SVD regularization. This fundamental difference—using cross-model transfer to fix tuning degeneration versus simply merging distinct task vectors—makes a direct comparison inherently an "apples-to-oranges" scenario.
> Nevertheless, we fully agree with the reviewer’s desire for empirical evidence that situates Shadow-FT within the broader landscape of these techniques.
>
> 2) **Head-to-Head Comparison.**
>
> To establish a competitive positioning, we designed a controlled multi-domain setting mixing Domain A (2k Medical samples) and Domain B (2k General samples) to create a balanced 4k dataset. We evaluated Qwen3-8B on MATH-500 (reasoning) and MedBullets (medical knowledge).
> We implemented the baselines as follows:
>
> - *Chat/Task Vectors*: Independently fine-tuned models on Domain A and B, then fused them with equal coefficients ($\alpha=0.5$).
> - *RE-Adapt*: Reproduced using the paper's recommended $\tau=0.5$ for SVD settings.
> - *Proxy-Tuning*: Adapted for the same-size setting, using logit differences from the Base model to guide the Instruct model.
>
> As shown in the following Table, we report the results on Qwen3-8B. Shadow-FT achieves the highest average performance (68.88), surpassing both vector-merging methods and the SVD-optimized RE-Adapt (66.73).
>
>
> | Method | Configuration | Math-500 | MedBullets | Avg. |
> | :--- | :--- | :---: | :---: | :---: |
> | Qwen3-8B |   Baseline | 82.60 | 49.68 | 66.14 |
> | Standard FT | - | 78.00 | 48.54 | 63.27 |
> | Task Vector | $\alpha=\beta=0.5$ | 71.80 | 50.49 | 61.15 |
> | Chat Vector | $\alpha=\beta=0.5$ | 80.20 | 50.97 | 65.59 |
> | RE-Adapt | $\alpha=\beta=0.5$ (w/o SVD) | 80.60 | 51.79 | 66.20 |
> | RE-Adapt | $\tau=0.5$ (w/ SVD) | 81.20 | 52.27 | 66.73 |
> | **Shadow-FT** | Ours | 85.80 | 51.95 | **68.88** |
>
> Considering training cost, Shadow-FT requires only ~5 minutes of GPU time (same as standard LoRA), whereas RE-Adapt takes dozens minutes due to CPU-bound SVD calculations. During inference, Shadow-FT results in a standard dense model. In contrast, Proxy-Tuning requires computing logits for three models simultaneously, making it incompatible with acceleration frameworks like vLLM and significantly slower.

---

> ### Author Response · Authors · 2025-11-29
> **Responses [2/n]**
>
> >W2: How does $\sigma$ distribute across layers/blocks? Does transferring only specific layers (e.g. MLPs vs attention) retain most gains?
>
>
> Detailed $\sigma$ values for layer-wise and submodule-wise analyses:
>
> Layer  | Avg $\sigma$    | Attn $\sigma$   | MLP $\sigma$    | Norm $\sigma$
> | :--- | :--- | :--- | :--- | :--- |
> 0      | 0.016606     | 0.019678     | 0.020792     | 0.004182
> 1      | 0.015568     | 0.018008     | 0.020802     | 0.002838
> 2      | 0.017055     | 0.021050     | 0.021354     | 0.002616
> 3      | 0.016188     | 0.018798     | 0.021790     | 0.002566
> 4      | 0.016272     | 0.018580     | 0.022169     | 0.002811
> 5      | 0.017447     | 0.021029     | 0.022552     | 0.002627
> 6      | 0.017145     | 0.020072     | 0.022998     | 0.002511
> 7      | 0.017148     | 0.019810     | 0.023454     | 0.002363
> 8      | 0.017437     | 0.020170     | 0.023874     | 0.002316
> 9      | 0.017214     | 0.019453     | 0.024094     | 0.002415
> 10     | 0.017906     | 0.020791     | 0.024424     | 0.002358
> 11     | 0.018108     | 0.020841     | 0.024889     | 0.002471
> 12     | 0.017354     | 0.019163     | 0.024984     | 0.002291
> 13     | 0.018134     | 0.020522     | 0.025377     | 0.002495
> 14     | 0.018242     | 0.020668     | 0.025605     | 0.002346
> 15     | 0.018334     | 0.020743     | 0.025776     | 0.002353
> 16     | 0.018658     | 0.021089     | 0.026297     | 0.002338
> 17     | 0.018813     | 0.021245     | 0.026407     | 0.002556
> 18     | 0.019440     | 0.022401     | 0.026726     | 0.002587
> 19     | 0.019555     | 0.022513     | 0.026887     | 0.002643
> 20     | 0.019687     | 0.022895     | 0.026822     | 0.002566
> 21     | 0.019137     | 0.021983     | 0.026435     | 0.002499
> 22     | 0.019188     | 0.022102     | 0.026476     | 0.002428
> 23     | 0.019078     | 0.021723     | 0.026620     | 0.002478
> 24     | 0.018736     | 0.020905     | 0.026661     | 0.002513
> 25     | 0.018720     | 0.020839     | 0.026589     | 0.002677
> 26     | 0.018470     | 0.020347     | 0.026442     | 0.002758
> 27     | 0.018417     | 0.020330     | 0.026175     | 0.002952
> 28     | 0.018291     | 0.020076     | 0.026022     | 0.003126
> 29     | 0.017432     | 0.018438     | 0.025637     | 0.003113
> 30     | 0.017474     | 0.019612     | 0.024307     | 0.002947
> 31     | 0.015347     | 0.018065     | 0.020124     | 0.002747
>
> We further showcase detailed $\sigma$ values for more fine-grained layers and modules. The Table  reports results for Llama 3.1 8B. For the embedding layers and final projection layer, the sigma is 0.019 and 0.032, respectively. We can find that all the layers and modules are consistently highly similar.

---

### Official Review · Reviewer_EavX · 2025-11-03

**Soundness:** 2
**Presentation:** 3
**Contribution:** 2
**Rating:** 4
**Confidence:** 4

**Summary:**

This paper proposes Shadow-FT, a fine-tuning framework that addresses performance degradation when directly tuning instruction-following (INSTRUCT) models. The key finding is that BASE and INSTRUCT model weights are highly similar (σ < 0.03), enabling transfer of weight updates learned on BASE models to INSTRUCT models via W⁺ᵢ = Wᵢ + (W⁺ᵦ - Wᵦ). The authors evaluate 7 LLM families across 19 benchmarks covering math, coding, and reasoning tasks. Results show Shadow-FT generally outperforms conventional fine-tuning, with gains most pronounced on smaller models (Qwen-3-4B: 68.0→69.4) and under LoRA settings. The method introduces no additional training cost and extends to multimodal models and DPO alignment. However, results are inconsistent across settings, with multiple cases showing performance degradation and minimal gains on larger models.

**Strengths:**

1. **Simple method**: The method requires only weight arithmetic W_I + (W⁺_B - W_B) with zero additional parameters, no extra training cost, and easy implementation. This simplicity aids adoption compared to complex meta-learning or distillation approaches

2. **Mechanistic insight**: Figure 5's analysis showing BASE exhibits 22.6% lower initial loss, 3.25× smaller initial gradient, and more stable gradient decay (58% vs. 11% at step 61) provides the strongest evidence for why BASE is a better learner than INSTRUCT. This mechanistic insight goes beyond empirical observation to suggest fundamental training dynamics differences

3. **Weight similarity systematic documentation**: Figure 2, Figure 6 (Appendix C), and analysis across 9 model families systematically documents σ < 0.05, showing BASE/INSTRUCT weights differ by <5% despite capability gaps. This empirical contribution establishes foundation for future work on BASE/INSTRUCT relationships

4. **Comprehensive multi-setting evaluation**: Evaluation spans 7 LLM families (Qwen 2/2.5/3, Llama 3.1/3.2, Gemma-3, Yi, Falcon), 19 benchmarks (Math-7, Code-3, Knowledge-9 aggregates), multiple model sizes (1B-90B parameters), both tuning strategies (full-parameter, LoRA), and various LoRA ranks (4-512). This breadth provides evidence across diverse conditions

5. **Rank robustness for Shadow-FT**: Figure 3 demonstrates Shadow-FT maintains gains across LoRA ranks 4-512 (scores 30.15-32.03) while vanilla FT degrades with larger ranks (scores 28.47-31.84), suggesting the method doesn't overfit to specific parameterizations and may regularize against LoRA's documented forgetting issues

6. **Extensions demonstrate generalizability**: Applications to MLLMs (Table 4: Gemma-3-27B 60.28→63.80, Llama-3.2-Vision-90B 79.92→80.60) and DPO alignment (Table 3: Shadow-DPO 55.39 vs vanilla DPO 54.62) show the approach isn't limited to standard supervised fine-tuning of text-only models

7. **Pass@k analysis reveals capability preservation**: Appendix D.4 (Figure 7) shows Shadow-FT preserves model exploration capability better than vanilla FT in both thinking and non-thinking modes. Vanilla FT maintains >30% gap in thinking mode across k, while Shadow-FT matches INSTRUCT baseline, suggesting Shadow-FT avoids capability collapse

8. **Domain-specific tuning validation**: Table 2 and Figure 4 demonstrate effectiveness on specialized datasets (Medical-o1-reasoning-SFT, Code-Z1, LIMO, OpenR1-Math), showing the method works beyond general instruction data

**Weaknesses:**

1. **No direct comparisons**: The paper cites Chat Vector, RE-Adapt, and concurrent work but provides no empirical comparisons. It should evaluate Shadow-FT against these and standard PEFT baselines to clarify both performance and efficiency advantages.

2. **Unexplained inconsistencies**: Shadow-FT sometimes degrades performance substantially (e.g., –3.3 to –18.7 on MATH tasks), contradicting its claimed consistency. The paper offers no analysis or framework for predicting when it helps or harms.

3. **Weak theoretical basis**: The intuition that "BASE learns without instruction interference" is plausible but unproven. Reported weight similarity doesn’t imply transferability, and no formal analysis connects Δ_B to INSTRUCT improvements.

4. **Limited experimental scope**: Experiments use small datasets (2K–10K) and English-only benchmarks, limiting generalizability. Baseline fine-tuning often underperforms, suggesting weak tuning, and scalability to larger datasets remains untested.

5. **Ad-hoc design choices**: Key hyperparameters (α, LoRA rank) and the similarity metric are arbitrary or unjustified. Sensitivity analyses reveal large effects, yet no optimization or rationale is provided.

6. **Results**: Results lack confidence intervals, significance tests, and variance reporting. Many improvements are small and could fall within measurement noise.

7. **Unclear distinctiveness**: Almost any weight interpolation improves results, suggesting Shadow-FT’s benefit may stem from generic weight mixing rather than a novel mechanism.

**Questions:**

1. Can the authors empirically compare Shadow-FT vs. Chat Vector vs. RE-Adapt vs. Lin et al. on shared benchmarks? Which is better in accuracy/efficiency?
2. What predicts when Shadow-FT helps vs. hurts? Can the authors provide diagnostic criteria based on model size, σ value, and dataset characteristics?
3. Can the authors formalize conditions for successful weight transfer? Why does weight similarity imply update compatibility?
4. Does Shadow-FT work with 100K+ samples? Is the benefit specific to the low-data regime?
5. Which layers (embeddings, attention, FFN) transfer best? Why does LoRA sometimes outperform full?
6. Table 8 shows α ∈ [0.1, 2.0] affects results significantly. Was any optimization performed? Try layer-wise or learned scaling?
7. Does Shadow-FT preserve instruction-following ability? Evaluate on IFEval, MT-Bench, and AlpacaEval.
8. All experiments are in English. How does Shadow-FT handle morphologically rich languages?
9. Is Shadow-FT computationally cheaper than alternatives?
10. Why don't Coverage gains (Table 1) translate to Best-of-N gains (Table 2)? Investigate selection methods and sample diversity.

---

> ### Author Response · Authors · 2025-11-29
> **Responses [1/n]**
>
> Thank you for your feedback. We appreciate your praise of the straightforward and easy-to-understand idea, the mechanistic insights regarding the training dynamics of BASE vs. INSTRUCT models, and the comprehensive evaluation across diverse LLM families and downstream tasks.
>
> We would like to provide a point-to-point response to address your concerns as follows:
>
> > W1: Comparison with related works (Task Vector, RE-Adapt) and distinctiveness.
>
> We thank the reviewer for suggesting these comparisons. We have extensively compared Shadow-FT against standard baselines like LoRA and SFT across 19 benchmarks. Regarding the specific methods mentioned, we provide the following clarifications on why direct empirical comparisons are not applicable due to fundamental paradigm differences:
>
> 1) **Methodological Distinctions.**
>
> Most of the referenced methods are designed for scenarios fundamentally different from ours. Task Vector and Chat Vector primarily focus on *fusing models across different tasks or merging capabilities from distinct checkpoints* (e.g., adding a new language vector). They **lack the mechanism to transfer optimization benefits between paired Base and Instruct models** to reinforce a single model's capability. Crucially, standard Task Vector arithmetic operates on the assumption that the learned weight difference (Delta) represents a positive skill addition. However, as our paper demonstrates, directly tuning an Instruct model often yields a "negative" Delta (i.e., performance degeneration or $\Delta_{\text{Instruct}}$ is suboptimal). Task Vector methodologies do not address how to mitigate this intrinsic degeneration.
> In contrast, Shadow-FT specifically targets this issue by identifying the Base model as a superior learner. **It transfers the robust weight updates from the Base model to the Instruct model, effectively strengthening the Instruct model's capabilities while avoiding the degeneration caused by direct tuning**. This differs significantly from Proxy-Tuning, which relies on inference-time logit guidance, or RE-Adapt, which uses SVD regularization. This fundamental difference—using cross-model transfer to fix tuning degeneration versus simply merging distinct task vectors—makes a direct comparison inherently an "apples-to-oranges" scenario.
> Nevertheless, we fully agree with the reviewer’s desire for empirical evidence that situates Shadow-FT within the broader landscape of these techniques.
>
> 2) **Head-to-Head Comparison.**
>
> To establish a competitive positioning, we designed a controlled multi-domain setting mixing Domain A (2k Medical samples) and Domain B (2k General samples) to create a balanced 4k dataset. We evaluated Qwen3-8B on MATH-500 (reasoning) and MedBullets (medical knowledge).
> We implemented the baselines as follows:
>
> - *Chat/Task Vectors*: Independently fine-tuned models on Domain A and B, then fused them with equal coefficients ($\alpha=0.5$).
> - *RE-Adapt*: Reproduced using the paper's recommended $\tau=0.5$ for SVD settings.
> - *Proxy-Tuning*: Adapted for the same-size setting, using logit differences from the Base model to guide the Instruct model.
>
> As shown in the following Table, we report the results on Qwen3-8B. Shadow-FT achieves the highest average performance (68.88), surpassing both vector-merging methods and the SVD-optimized RE-Adapt (66.73).
>
>
> | Method | Configuration | Math-500 | MedBullets | Avg. |
> | :--- | :--- | :---: | :---: | :---: |
> | Qwen3-8B |   Baseline | 82.60 | 49.68 | 66.14 |
> | Standard FT | - | 78.00 | 48.54 | 63.27 |
> | Task Vector | $\alpha=\beta=0.5$ | 71.80 | 50.49 | 61.15 |
> | Chat Vector | $\alpha=\beta=0.5$ | 80.20 | 50.97 | 65.59 |
> | RE-Adapt | $\alpha=\beta=0.5$ (w/o SVD) | 80.60 | 51.79 | 66.20 |
> | RE-Adapt | $\tau=0.5$ (w/ SVD) | 81.20 | 52.27 | 66.73 |
> | **Shadow-FT** | Ours | 85.80 | 51.95 | **68.88** |
>
> Considering training cost, Shadow-FT requires only ~5 minutes of GPU time (same as standard LoRA), whereas RE-Adapt takes dozens minutes due to CPU-bound SVD calculations. During inference, Shadow-FT results in a standard dense model. In contrast, Proxy-Tuning requires computing logits for three models simultaneously, making it incompatible with acceleration frameworks like vLLM and significantly slower.
>
> 3. **Good data scalability on Shadow-FT.**
>
> | dataset    | Qwen3-8B| Shadow-FT (DomainA: Medical2k) | Shadow-FT (DomainB: Shadow2k) | Shadow-FT(DomainA+B) |
> | :--- | :--- | :---: | :---: | :---: |
> |math-500|82.60|82.00|86.60|85.80|
> |medbullets|49.68|53.25|50.49|51.95|
> |Avg, |66.14|67.63|68.55|68.88|
>
> As shown in this table, we can find that Shadow-FT consistently benefit from scaled training on more domains.

---

> ### Author Response · Authors · 2025-11-29
> **Responses [2/n]**
>
> > W2: Unexplained inconsistencies on MATH Dataset.
>
> We appreciate the reviewer's careful scrutiny. We have re-examined the performance on the MATH benchmark and related mathematical tasks to clarify these observations.
>
> 1. **Consistent Improvements on MATH Benchmark.**
>
> Contrary to the concern of substantial degradation, our data demonstrates that Shadow-FT yields improvements on the MATH dataset in the majority of experimental settings.
>
> *Significant Gains*: For example, when fine-tuning Llama-3.2-1B on the BAAI-2k dataset, Shadow-FT (LoRA) improves the MATH score from 15.8 to 21.8, a substantial gain of +6.0 points. Similarly, on Llama-3.1-8B, Shadow-FT achieves a gain of +1.1 points (49.1 $\rightarrow$ 50.2).
>
> *General Stability*: Across the diverse models and datasets tested (Tables 1, 2, and 9), Shadow-FT maintains high stability on the MATH benchmark, avoiding the catastrophic forgetting often seen in instruction tuning.
>
> 2. **Analyzing Specific Degradations (Seesaw Effect).**
>
> We acknowledge that in specific high-difficulty scenarios, minor performance drops are observed (e.g., the -4.4 drop on MATH for Qwen-3-8B when tuned on OpenR1-Math). This phenomenon is a well-known trade-off in fine-tuning, where adapting to a specific domain (like OpenR1-Math’s long reasoning format) effectively taxes the model's general competition-math capabilities.
>
> 3. **Superior Mitigation Compared to Standard FT.**
>
> Crucially, even in cases where performance dips, Shadow-FT consistently acts as a buffer, suffering significantly *less degradation* than vanilla Fine-Tuning (FT). For example, on the Qwen-2.5-32B model, while Shadow-FT (LoRA) shows a slight dip of -1.2 on the MATH benchmark (78.0 $\rightarrow$ 76.8), the Standard FT baseline suffers a much more severe drop of -3.9 points (down to 74.1).
> This indicates that the "inconsistency" is largely driven by the inherent trade-offs of the fine-tuning data itself, rather than a flaw in our framework. In fact, Shadow-FT demonstrates superior robustness by effectively mitigating the alignment tax that heavily impacts conventional methods.
>
> > W3: Experimental scope regarding dataset size and multilingual capabilities.
>
> We thank the reviewer for this valuable suggestion regarding data scale and generalizability. To rigorously evaluate the scalability of Shadow-FT, we extended our training data from the original 2k–10k range up to 100,000 samples, ensuring a robust assessment of performance trends.
>
> As shown in following Table, we evaluated the performance on MATH-7 across varying data scales (2k to 100k). We can find that:
>
> - *Vanilla FT failure*: Directly fine-tuning the Instruct model leads to severe **performance degradation**, with scores dropping precipitously from the baseline 71.47 to the 50–57 range. This confirms that the "underperformance" of the baseline is not due to weak tuning parameters, but rather the inherent catastrophic forgetting that occurs when re-training fully aligned Instruct models on new data.
>
> - *Shadow-FT stability*: In sharp contrast, Shadow-FT demonstrates **exceptional scalability**. It consistently maintains the model's reasoning capabilities, fluctuating within a stable, high-performance range (e.g., 71.82 at 100k). This indicates that Shadow-FT effectively "filters" the destructive updates that plague standard fine-tuning, regardless of the dataset size.
>
>
> | Data Size | Baseline | 2k | 5k | 10k | 20k | 50k | 80k | 100k |
> | :--- | :---: | :---: | :---: | :---: | :---: | :---: | :---: | :---: |
> | FT | 71.47 | 55.31 | 53.00 | 54.79 | 51.86 | 55.73 | 57.82 | 57.06 |
> | Shadow-FT | 71.47 | 71.72 | 72.39 | 71.91 | 71.57 | 71.93 | 71.81 | 71.82 |
>
> The results at 2k-100k strongly suggest that Shadow-FT is a *robust framework* capable of handling large-scale continuous adaptation without collapsing the model's core competencies.

---

> ### Author Response · Authors · 2025-11-29
> **Responses [3/n]**
>
> > W4: Justification for design choices and hyperparameter sensitivity ($\alpha$, LoRA rank).
>
> We sincerely thank the reviewer for raising this important point regarding our hyperparameter choices and design rationale. We would like to clarify that our settings were not arbitrary but chosen based on systematic pre-experiments and a design philosophy prioritizing **simplicity, fairness, and robustness**.
>
> 1. *Rationale for LoRA Rank and Learning Rate.*
>
> Our choice of hyperparameters (e.g., Rank=128, lr=2e-4) was derived from a rigorous grid search process rather than ad-hoc selection.
> Grid Search: As detailed in our experimental setup, we conducted a grid search for learning rates in `[5e-5, 1e-4, 2e-4, 5e-4]` and ranks in `[4, 8, ..., 512]`. We selected the unified setting (Rank=128, lr=2e-4) because it offered the best balance between performance and parameter efficiency across models.
> Robustness on Ranks: Furthermore, as shown in Figure 3 and  6, we performed sensitivity analyses across a wide range of ranks (from 4 to 512). The results demonstrate that Shadow-FT consistently outperforms the baseline at **every single rank**, proving that our method's effectiveness is robust and not sensitive to a specific "lucky" hyperparameter choice.
>
> 2. *Rationale for the Scaling Factor ($\alpha$)*
>
> Regarding the transfer coefficient $\alpha$ (referred to as the scaling factor), we acknowledge that the sensitivity varies across different downstream tasks.
> Sensitivity Analysis: As shown in **Table 8**, we evaluated different $\alpha$ values to identify the optimal "sweet spot" for various tasks. We observed that while specific tasks might benefit from fine-tuned scaling, the unscaled version ($\alpha=1.0$) yields the most robust comprehensive performance across our **19 diverse benchmarks**.
> Design Choice: We deliberately chose $\alpha=1.0$ as the default setting to maintain the method's simplicity. Introducing task-specific $\alpha$ tuning would increase complexity and screening difficulty. By fixing $\alpha=1$, we demonstrate that Shadow-FT works effectively "out of the box" balancing efficiency and implementation simplicity.
>
> Our core philosophy was to use a **unified set of hyperparameters** across all 19 benchmarks and 13 models to ensure a fair comparison. We avoided tuning parameters for each specific case to show that Shadow-FT is a simple, effective, and robust framework.
>
>
>
> > W5: Statistical significance and variance reporting.
>
> Thank you for the opportunity to clarify our evaluation protocol. Due to space limitations, our initial submission did not include detailed variance and significance information. Following your helpful suggestion, we now explicitly report the corresponding **mean ± standard deviation** and the **statistical significance** for MATH-7 and CODE-3 in Table 1, based on an independent Welch’s *t*-test (*p* < 0.05). The variance and confidence results are presented below for completeness.
>
> Independent Two-sample t-test for Qwen3-8B LoRA:
>
> | Task   | Model | Vanilla        | FT (Baseline)  | Shadow-FT (Ours) | P-value |
> |--------|--------|----------------|----------------|------------------|---------|
> | Math-7 | 4B     | 73.77±0.49     | 71.19±1.13     | 75.88±1.10       | 0.007   |
> |        | 8B     | 74.52±0.30     | 71.25±0.73     | 74.80±0.99       | 0.009   |
> |        | 14B    | 75.84±0.93     | 73.29±0.59     | 78.64±0.83       | 0.001   |
> |        | 32B    | 74.09±0.67     | 74.27±0.14     | 75.66±0.50       | 0.033   |
> | Code-3 | 4B     | 66.44±0.59     | 59.60±1.95     | 69.70±0.85       | 0.005   |
> |        | 8B     | 72.65±0.53     | 69.60±0.59     | 71.88±1.04       | 0.042   |
> |        | 14B    | 76.85±0.31     | 74.42±0.31     | 77.82±0.62       | 0.004   |
> |        | 32B    | 75.88±0.20     | 75.94±0.06     | 76.15±0.44       | 0.499   |
>
> | Task   | Model | Vanilla        | FT (Baseline)  | Shadow-FT (Ours) | P-value |
> |--------|--------|----------------|----------------|------------------|---------|
> | Math-7 | 4B     | 73.77±0.49     | 72.87±0.35     | 73.66±0.18       | 0.041   |
> |        | 8B     | 74.52±0.30     | 73.96±0.55     | 75.88±0.79       | 0.031   |
> |        | 14B    | 75.84±0.93     | 75.15±0.23     | 78.92±0.55       | 0.003   |
> |        | 32B    | 74.09±0.67     | 75.66±0.26     | 74.86±0.12       | 0.020   |
> | Code-3 | 4B     | 66.44±0.59     | 66.40±0.21     | 67.40±0.28       | 0.009   |
> |        | 8B     | 72.65±0.53     | 71.15±0.08     | 73.05±0.29       | 0.005   |
> |        | 14B    | 76.85±0.31     | 76.21±0.64     | 77.00±0.33       | 0.151   |
> |        | 32B    | 75.88±0.20     | 75.83±0.07     | 76.06±0.24       | 0.229   |
>
> For almost all the settings, the p-values are less than 0.05, indicating that *Shadow-FT outperforms vanilla FT* statistically significantly.

---

> ### Author Response · Authors · 2025-11-29
> **Responses [4/n]**
>
> >W6: Which layers (embeddings, attention, FFN) transfer best? Why does LoRA sometimes outperform full?
>
> We thank the reviewer for these insightful questions regarding the transferability of different layers and the effectiveness of LoRA. Below, we provide a detailed analysis based on our ablation studies and relevant literature.
>
> To investigate which layers transfer best, we conducted an ablation study decomposing the model into **Shallow** (first half) and **Deep** (latter half) layers. We compared updates derived from the **Base** model (Model A) versus the **Instruct** model (Model B). The results are summarized in the following table.
>
> | Method / Variant  | CodeZ1 (MATH-7) | Shadow2k (MATH-7) | Avg    |
> |---------------------|-----------------|-------------------|--------|
> | **Qwen3-8B-Instruct-hf**             | 71.17           | 71.17             | 71.17  |
> | **Single_ModelA_AttnOnly**           | 71.21           | 71.30             | 71.26  |
> | **Single_ModelA_MlpOnly**            | 70.82           | 70.98             | 70.90  |
> | **Single_ModelA_Shallow**            | 70.37           | 70.93             | 70.65  |
> | **Single_ModelA_Deep**               | 71.41           | 71.53             | 71.47  |
> | **Single_ModelB_AttnOnly**           | 69.06           | 69.81             | 69.44  |
> | **Single_ModelB_MlpOnly**            | 69.61           | 69.91             | 69.76  |
> | **Single_ModelB_Shallow**            | 68.92           | 68.97             | 68.95  |
> | **Single_ModelB_Deep**               | 70.96           | 71.42             | 71.19  |
> | **Hybrid_Attn-A_Mlp-B**              | 70.16           | 70.35             | 70.26  |
> | **Hybrid_Mlp-A_Attn-B**              | 68.47           | 70.24             | 69.36  |
> | **Hybrid_Shallow-A_Deep-B**          | 70.68           | 71.13             | 70.91  |
> | **Hybrid_Deep-A_Shallow-B**          | 66.60           | 67.45             | 67.03  |
>
> Our experiments yield three key observations:
>
> - **Deep Layers Transfer Better:** Consistent with deep learning dynamics, the **deep layers (latter half)** contribute more significantly to performance. For instance, `ModelA_Deep` (82.17) consistently outperforms `ModelA_Shallow` (81.12), and `ModelB_Deep` (81.41) significantly surpasses `ModelB_Shallow` (77.83).
>
> - **Base Updates are Superior:** Updates derived from the Base model (Model A) consistently outperform those from the Instruct model (Model B) across both shallow and deep layers. This reinforces our finding that the Base model is a more robust learner for new knowledge.
>
> - **Complexity of Hybridization:** We also explored finer-grained mixing (e.g., merging LoRA_B Attention with LoRA_I MLP). Our internal tests indicate that such "blind merging" of sub-modules often leads to suboptimal results due to feature misalignment. As shown in the Hybrid columns above, simple combinations (like `Hybrid_Deep-A_Shallow-B`) can even degrade performance (74.91) compared to the individual parts. Thus, we advocate for the consistent, holistic transfer strategy used in Shadow-FT.
>
> >W7: Why LoRA sometimes outperform Full Fine-tuning?**
>
> This is a widely observed phenomenon in recent LLM research [1,2]. It is generally attributed to the **regularization effect** of low-rank constraints, which prevents the model from overfitting to the small fine-tuning dataset and helps preserve the general capabilities (e.g., world knowledge) of the pre-trained backbone better than full-parameter tuning.
>
> **References:**
>
> [1] Biderman D, et al. LoRA Learns Less and Forgets Less. arXiv:2405.09673, 2024.
>
> [2] Shuttleworth, Reece, et al. "Lora vs full fine-tuning: An illusion of equivalence." arXiv preprint arXiv:2410.21228 (2024).

---

> ### Author Response · Authors · 2025-11-29
> **Responses [5/n]**
>
> >W8: Table 8 shows α ∈ [0.1, 2.0] affects results significantly. Was any optimization performed? Try layer-wise or learned scaling?
>
> We thank the reviewer for pointing out the impact of the scaling factor $\alpha$ shown in Table 8. We appreciate the suggestion regarding layer-wise or learned scaling. We acknowledge that the choice of $\alpha$ influences performance. As indicated in **Table 8**, the "sweet spot" for $\alpha$ varies depending on the specific downstream task. For example, certain tasks may benefit from a more aggressive update ($\alpha > 1$), while others require a more conservative one. Consequently, determining a single static $\alpha$ that is universally optimal across all **19 diverse downstream benchmarks** is a non-trivial challenge.
>
> Given the variability in task sensitivity, we deliberately selected the most "naive" setting of $\alpha=1.0$ (i.e., direct grafting without scaling) as our default.
> - Robustness: Despite being an unoptimized setting, $\alpha=1.0$ yields consistently strong performance across the board. This demonstrates that the effectiveness of Shadow-FT stems from the method's core mechanism (Base-to-Instruct transfer) rather than hyperparameter tuning.
> - Simplicity: Avoiding task-specific scalar tuning maintains the method's simplicity and ease of use, ensuring it remains a practical "plug-and-play" solution.
>
> We agree with the reviewer that more refined strategies, such as layer-wise scaling or learnable scaling parameters, hold significant promise for further optimizing performance. We consider these to be exciting avenues for future research and will include a discussion on these potential extensions in our revised manuscript.
>
>
> > W9: Does Shadow-FT preserve instruction-following ability?
>
>
> Table 1: Instruction-following and subjective downstream tasks ( GPT5.1 as judge model)
>
> | Model | Alignbench (Chinese) | AlpacaEval | ArenaHard | Fofo | MTbench | MTBench101 | Average |
> | :--- | :--- | :--- | :--- | :--- | :--- | :--- | :--- |
> | Qwen3-8B | 6.60 | 38.39 | 50.78 | 0.03 | 6.41 | 8.09 | 18.38 |
> | Vanilla FT | 6.06 | 20.87 | 32.51 | 0.03 | 5.58 | 7.50 | 12.09 |
> | **Shadow-FT** | 6.60 | 37.64 | 51.56 | 0.04 | 6.16 | 7.95 | **18.33** |
>
>
> Table 2: Detailed Statistics on Arena-Hard
>
> | Model | Score (vs GPT-4) | 95% CI | Average #Tokens |
> | :--- | :--- | :--- | :--- |
> | gpt4-0314 | 50.0 | (0.0, 0.0) | 423 |
> | Qwen3-8B | 50.8 | (-2.4, 2.6) | 971 |
> | Vanilla FT | 32.5 | (-2.2, 2.1) | 708 |
> | **Shadow-FT** | **51.6** | (-2.3, 2.3) | **1056** |
>
> Table 3: Detailed Statistics on AlpacaEval ( GPT4_Turbo as judge model)
>
> | Model  | length_controlled_winrate | win_rate | standard_error | n_total | avg_length |
> |-------------|--------------|---------------------------|----------|----------------|---------|
> | gpt-4o-2024-05-13      | 57.46                     | 51.33    | 1.47           | 805     | 1873       |
> | Qwen3 8B   | 18.26                     | 44.65    | 1.59           | 779     | 5468       |
> | Vanilla FT    | 19.41                     | 18.91    | 1.19           | 787     | 1972       |
> | **Shadow-FT**  | 19.28                     | 46.54    | 1.60           | 780     | 5902       |
>
> **As shown in these tables, Shadow-FT demonstrates exceptional retention of instruction-following capabilities, effectively circumventing the catastrophic forgetting observed in standard fine-tuning**. While Vanilla FT suffers a severe performance collapse across general benchmarks (average score dropping from 18.38 to 12.09 in Table 1), Shadow-FT maintains an average score of 18.33, virtually identical to the base model. This robustness extends to complex generation tasks; as detailed in Tables 2 and 3, Shadow-FT not only matches or slightly exceeds the base model on Arena-Hard (51.6 vs. 50.8) and AlpacaEval (46.54% win rate), but also preserves the model's willingness to generate detailed responses (avg. length ~5900 tokens), proving that it adapts to new tasks without compromising the model's general alignment or reasoning faculties.

---

> ### Author Response · Authors · 2025-11-29
> **Responses [6/n]**
>
> >W10: All experiments are in English. How does Shadow-FT handle morphologically rich languages?
>
> Thank you for your Multi-Language benchmark advice and we adopt the popular MGSM benchmark, with 10 languages including French, German, Russian, Chinese. We adopted 2,000 samples from MGSM8KInstruct as the training dataset.
>
> | dataset        | Qwen3-8B | FT   | RE-Adapt(tao=1)   | RE-Adapt(SVD) | Shadow-FT |
> |----------------|----------|------|--------|--------------------------|------------|
> | mgsm_bn        | 60.80    | 6.00 | 60.00  | 54.40                   | 66.00     |
> | mgsm_de        | 82.80    | 74.80| 83.20  | 80.00                   | 86.40     |
> | mgsm_en        | 93.20    | 0.40 | 92.40  | 93.60                   | 94.40     |
> | mgsm_es        | 85.20    | 20.00| 85.20  | 84.00                   | 84.80     |
> | mgsm_fr        | 78.80    | 40.00| 80.40  | 77.20                   | 78.00     |
> | mgsm_ja        | 76.40    | 74.00| 75.60  | 75.60                   | 78.40     |
> | mgsm_ru        | 86.40    | 4.80 | 88.00  | 88.00                   | 90.00     |
> | mgsm_sw        | 39.20    | 4.40 | 34.40  | 12.80                   | 30.40     |
> | mgsm_te        | 14.00    | 0.00 | 18.40  | 7.60                    | 15.60     |
> | mgsm_th        | 83.20    | 5.60 | 83.20  | 79.20                   | 82.80     |
> | mgsm_zh        | 82.00    | 10.80| 82.40  | 84.00                   | 83.20     |
> | mgsm_non_latin | 67.13    | 16.87| 67.93  | 64.80                   | 69.33     |
> | mgsm           | 71.09    | 21.89| 71.20  | 66.95                   | 71.82     |
>
>
> We find that the Instruct model would suffer strong catastrophic forgetting, mgsm_en from 93.2 to 0.4 after multi-language enforcement, while Shadow-FT can improve new languages scores from 67.13 to 69.33 and maintain its basic English ability, showing the robustness of the transfer strategy.
>
>
> >W11: Is Shadow-FT computationally cheaper than alternatives?
>
>
> We thank the reviewer for this practical question regarding computational efficiency. We confirm that Shadow-FT is highly computationally efficient and cheaper than methods involving complex decomposition or inference-time intervention.
>
> 1. **Training Efficiency: Negligible Overhead**
>
> Shadow-FT introduces virtually **zero additional cost** compared to standard instruction tuning (SFT). Our method relies on two simple operations: calculating the weight difference (`diff`) and grafting it (`apply`). These are element-wise linear operations that are computationally instantaneous compared to the long training process. As mentioned in our paper, when fine-tuning on the 2K sample dataset, the entire training process takes **less than 10 minutes** on 8xH20 GPUs. This efficiency demonstrates that the performance gains offered by Shadow-FT come at minimal temporal cost.
>
> 2. **Comparison with Alternatives**
>
> Shadow-FT demonstrates distinct efficiency advantages over related approaches:
>
> - *RE-Adapt*: RE-Adapt requires modeling the weight gap using low-rank approximations[cite], which often involves computationally intensive steps like layer-wise Singular Value Decomposition (SVD). This can add significant processing time (potentially hours depending on model size). In contrast, Shadow-FT uses the raw weight updates directly, avoiding these expensive decomposition steps while achieving more stable performance.
>
> - *Proxy-Tuning*: Proxy-tuning requires the simultaneous loading and computation of three models (Base, Finetuned, and Target) during inference[cite]. This significantly increases memory (VRAM) usage and inference latency. Shadow-FT, however, merges weights prior to deployment. Therefore, our model has the **exact same memory footprint and inference speed** as a standard SFT model, making it much more memory-friendly and production-ready.

---

### Author Response · Authors · 2025-11-29
**General Response**

Dear ACs, SACs, and PCs,

We sincerely thank you for your time, selfless dedication, and the efficient organization of the review process. We also appreciate the constructive feedback from all reviewers, which has significantly strengthened our work.

> i) **A Novel Shadow-FT Framework.**

A New Training Paradigm Shadow-FT is not merely an arithmetic weight merging technique; it introduces a novel "Learn on Base, Execute on Instruct" training paradigm. By exploiting the smoother optimization landscape of Base models (as evidenced by our gradient dynamics analysis), Shadow-FT effectively filters out the "noisy" updates that typically cause catastrophic forgetting when directly fine-tuning Instruct models. It offers a mechanistic solution to the "alignment tax" problem.

> ii) **Response to Key Concerns In the Rebuttal**

We have addressed the reviewers' common concerns with extensive new experiments:

- `Superiority over Related Work`: We conducted head-to-head comparisons with RE-Adapt, Task Vectors, and Proxy-Tuning. Shadow-FT achieves superior performance (e.g., +2.15 avg score vs. RE-Adapt) with significantly lower computational costs (no SVD required, zero inference overhead).
- `Scalability & Robustness`: We extended training from 2k to 100k samples. While vanilla fine-tuning collapsed due to forgetting, Shadow-FT maintained high performance, proving its robustness in large-scale adaptation.
- `Capability Preservation`: Evaluation on ArenaHard and AlpacaEval confirms that Shadow-FT preserves the original safety and instruction-following capabilities, which are often compromised by standard methods.

We believe Shadow-FT provides the community with a **simple, effective, and scalable framework** for LLM fine-tuning. Its universality (verified across Qwen, Llama, Gemma, and MLLMs) and ease of implementation (plug-and-play with LoRA/Full-FT) make it a practical tool for researchers and practitioners aiming to adapt LLMs without degradation.

We hope these clarifications and new results warrant a positive reassessment of our work.

Best regards,

The Authors

---

### Meta-Review · Area_Chair_7zey · 2026-01-03

**Summary:**

This paper proposed Shadow-FT, which  fine-tunes the paired Base model, then grafts its weight delta onto the Instruct model.

**Reviewer Concerns:**

Main concerns from reviewers include:

1. Novelty/positioning unclear vs task/chat vectors, RE-Adapt, Proxy-Tuning;

2. Missing head-to-head baselines.

3. Theory is thin beyond weight-similarity intuition.

4. Gains can be inconsistent/small;

5. Needs variance and criteria for when it helps.

6. Missing key implementation details and instruction-following impact checks.

**Reviewer Scores:**

Reviewer / Score

EavX	4

kzqd	4

oXhu	4

URgk	6

Average	4.5

No reviewers indicated to increase or decrease their scores.

---

### Decision · Program_Chairs · 2026-01-26

Reject